# EXPLORING TRANSFORMER BACKBONES FOR HETEROGENEOUS TREATMENT EFFECT ESTIMATION

## ABSTRACT

Previous works on Treatment Effect Estimation (TEE) are not in widespread use because they are predominantly theoretical, where strong parametric assumptions are made but untractable for practical application. Recent works use Multilayer Perceptron (MLP) for modeling casual relationships, however, MLPs lag far behind recent advances in ML methodology, which limits their applicability and generalizability. To extend beyond the single domain formulation and towards more realistic learning scenarios, we explore model design spaces beyond MLPs, i.e., transformer backbones, which provide flexibility where attention layers govern interactions among treatments and covariates to exploit structural similarities of potential outcomes for confounding control. Through careful model design, **Trans**formers as **T**reatment **E**ffect **E**stimators (TransTEE) is proposed. We show empirically that TransTEE can: (1) serve as a general-purpose treatment effect estimator which significantly outperforms competitive baselines on a variety of challenging TEE problems (e.g., discrete, continuous, structured, or dosage-associated treatments.) and is applicable to both when covariates are tabular and when they consist of structural data (e.g., texts, graphs); (2) yield multiple advantages: compatibility with propensity score modeling, parameter efficiency, robustness to continuous treatment value distribution shifts, explainable in covariate adjustment, and real-world utility in auditing pre-trained language models.

## 1 INTRODUCTION

One of the fundamental tasks in causal inference is to estimate treatment effects given covariates, treatments and outcomes. Treatment effect estimation is a central problem of interest in clinical healthcare and social science (Imbens & Rubin, 2015), as well as econometrics (Wooldridge, 2015). Under certain conditions (Rosenbaum & Rubin, 1983), the task can be framed as a particular type of missing data problem, whose structure is fundamentally different in key ways from supervised learning and entails a more complex set of covariate and treatment representation choices.

Previous works in statistics leverage parametric models (Imbens & Rubin, 2015; Wager & Athey, 2018; Künzel et al., 2019; Foster & Syrgkanis, 2019) to estimate heterogeneous treatment effects. To improve their utilities, feed-forward neural networks have been adapted for modeling causal relationships and estimating treatment effects (Yoon et al., 2018; Bica et al., 2020b; Schwab et al., 2020; Nie et al., 2021; Curth & van der Schaar, 2021b), in part due to their flexibility in modeling nonlinear functions (Hornik et al., 1989) and high-dimensional input (Johansson et al., 2016). Among them, the specialized NN's architecture plays a key role in learning representations for counterfactual inference (Alaa & Schaar, 2018; Curth & van der Schaar, 2021b) such that treatment variables and covariates are well distinguished (Shalit et al., 2017).

Despite these encouraging results, several key challenges make it difficult to adopt these methods as standard tools for treatment effect estimation. Most current works based on subnetworks do not sufficiently exploit the structural similarities of potential outcomes for heterogeneous TEE[1] and accounting for them needs complicated regularizations, reparametrization or multi-task architectures that are problem-specific (Curth & van der Schaar, 2021b). Moreover, they heavily rely on their treatment-specific designs and cannot be easily extended beyond the narrow context in which they are originally. For example, they have poor practicality and generalizability when high-dimensional

---

[1]For example, $\mathbb{E}[Y(1) - Y(0)|X]$ is often of a much simpler form to estimate than either $\mathbb{E}[Y(1)|X]$ or $\mathbb{E}[Y(0)|X]$, due to inherent similarities between $Y(1)$ and $Y(0)$.

Table 1: **Comparison of existing works and TransTEE in terms of parameter complexity.** $n$ is the number of treatments. $B_T, B_D$ are the number of branches for approximating continuous treatment and dosage. Treatment interaction means explicitly modeling collective effects of multiple treatments. TransTEE is general for all the factors.

| METHODS | DISCRETE TREATMENT | CONTINUOUS TREATMENT | TREATMENT INTERACTION | DOSAGE |
|---|---|---|---|---|
| TARNET (SHALIT ET AL., 2017) | $\mathcal{O}(n)$ | | | |
| PERFECT MATCH (SCHWAB ET AL., 2018) | $\mathcal{O}(n)$ | | $\mathcal{O}(2^T)$ | |
| DRAGONNET (SHI ET AL., 2019) | $\mathcal{O}(n)$ | | | |
| DRNET (SCHWAB ET AL., 2020) | $\mathcal{O}(n)$ | | | $\mathcal{O}(TB_D)$ |
| SCIGAN (BICA ET AL., 2020B) | $\mathcal{O}(n)$ | | | $\mathcal{O}(TB_D)$ |
| VCNET (NIE ET AL., 2021) | $\mathcal{O}(1)$ | $\mathcal{O}(1)$ | | |
| NCORE (PARBHOO ET AL., 2021) | $\mathcal{O}(n)$ | $\mathcal{O}(B_T)$ | $\mathcal{O}(n)$ | |
| FLEXTENET (CURTH & VAN DER SCHAAR, 2021B) | $\mathcal{O}(n)$ | | | |
| OURS | $\mathcal{O}(1)$ | $\mathcal{O}(1)$ | $\mathcal{O}(1)$ | $\mathcal{O}(1)$ |

structural data (e.g., texts and graphs) are given as input (Kaddour et al., 2021). Besides, those MLP-based models currently lag far behind recent advances in machine learning methodology, which are prone to issues of scale, expressivity and flexibility. Specifically, those side limitations include parameter inefficiency (Table 1), and brittleness under different scenarios, such as when treatments shift slightly from the training distribution. The above limitations clearly show a pressing need for an effective and practical framework to estimate treatment effects.

In this work, we explore recent advanced models in the deep learning community to boost the model design for TEE tasks. Specifically, the core idea of our approach consists of three parts: as an S-learner, TransTEE embeds all treatments and covariates, which avoids multi-task architecture and shows improved flexibility and robustness to continuous treatment value distribution shifts; attention mechanisms are used for modeling treatment interaction and treatment-covariate interaction. In this way, TransTEE enables adaptive covariate selection (De Luna et al., 2011; VanderWeele, 2019) for inferring causal effects. For example, one can observe in Figure 1 that both pre-treatment covariates and confounders are appropriately adjusted with higher

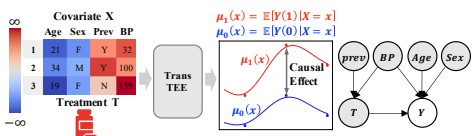

Figure 1: **A motivating example** with a corresponding causal graph. **Prev** denotes previous infection condition and **BP** denotes blood pressure. TransTEE adjusts an appropriate covariate set $\{\textbf{Prev}, \textbf{BP}\}$ with attention which is visualized via a heatmap.

weights, which recovers the "disjunctive cause criterion" (De Luna et al., 2011) that accounts for those two kinds of covariates and is helpful for ensuring the plausibility of the conditional ignorability assumption when complete knowledge of a causal graph is not available. This recipe also gives improved versatility when working with heterogeneous treatments types (Figure 2).

Our first contribution shows that transformer backbones, equipped with proper design choices, can be effective and versatile treatment effect estimators under the Rubin-Neyman potential outcomes framework. TransTEE is empirically verified to be (i) a flexible framework applicable for a wide range of TEE settings; (ii) compatible and effective with propensity score modeling; (iii) parameter-efficient; (iv) explainable in covariate adjustment; (v) robust under continuous treatment shifts; (vi) useful for debugging pre-trained language models (LMs) to promote favorable social outcomes.

Moreover, comprehensive experiments on six benchmarks with four types of treatments are conducted to verify the effectiveness of TransTEE in estimating treatment effects. We show that TransTEE produces covariate adjustment interpretation and significant performance gains given discrete, continuous or structured treatments on popular benchmarks including IHDP, News, TCGA. We introduce a new surrogate modeling task to broaden the scope of TEE beyond semi-synthetic evaluation and show that TransTEE is effective in real-world applications like auditing fair predictions of LMs.

## 2 RELATED WORK

**Neural Treatment Effect Estimation.** There are many recent works on adapting neural networks to learn counterfactual representations for treatment effect estimation (Johansson et al., 2016; Shalit et al., 2017; Louizos et al., 2017; Yoon et al., 2018; Bica et al., 2020b; Schwab et al., 2020; Nie et al.,

2021; Curth & van der Schaar, 2021b). To mitigate the imbalance of covariate representations across treatment groups, various approaches are proposed including optimizing distributional divergence (e.g. IPM including MMD, Wasserstein distance), entropy balancing (Zeng et al., 2020) (converges to JSD between groups), counterfactual variance (Zhang et al., 2020). However, their domain-specific designs make them limited to different treatments as shown in Table 1: methods like VCNet (Nie et al., 2021) use a hand-crafted way to map a real-value treatment to an $n$-dimension vector with a constant mapping function, which is hard to converge under shifts of treatments (Table 6 in Appendix); models like TARNet (Shalit et al., 2017) need an accurate estimation of the value interval of treatments. Moreover, previous estimators embed covariates to only one representation space by fully connected layers, tending to lose their connection and interactions (Shalit et al., 2017; Johansson et al., 2020). And it is non-trivial to adapt to the wider settings given existing ad hoc designs on network architectures. For example, the case with $n$ treatments and $m$ associated dosage requires $n \times m$ branches for methods like DRNet (Schwab et al., 2020), which put a rigid requirement on the extrapolation capacity and infeasible given observational data.

**Transformers and Attention Mechanisms** Transformers (Vaswani et al., 2017) have demonstrated exemplary performance on a broad range of language tasks and their variants have been successfully adapted to representation learning over images (Dosovitskiy et al., 2021), programming languages (Chen et al., 2021), and graphs (Ying et al., 2021) partly due to their flexibility and expressiveness. Their wide utility has motivated a line of work for general-purpose neural architectures (Jaegle et al., 2021; 2022) that can be trained to perform tasks across various modalities like images, point clouds, audios and videos. But causal inference is fundamentally different from the above models' focus, i.e. supervised learning. And one of our goals is to explore the generalizability of attention-based models for TEE across domains with high-dimensional inputs, an important desideratum in causal representation learning (Schölkopf et al., 2021). There are recent attempts to use attention mechanisms for TEE Tasks (Guo et al., 2021; Xu et al., 2022). CETransformer (Guo et al., 2021) uses embeds covariates for different treatments as a T-learner, They only trivially learn covariate embeddings but not treatment embedding, while the latter is shown more important for TEE tasks. In contrast, TransTEE is an S-learner, which is more well-suited to account for causal heterogeneity (Künzel et al., 2019; Curth & van der Schaar, 2021b;a). ANU (Xu et al., 2022) utilizes attention mechanisms to map the original covariate space $X$ into a latent space $Z$ with a single model. We detail the difference in Appendix A.

## 3 PROBLEM STATEMENT AND ASSUMPTIONS

**Treatment Effect Estimation.** We consider a setting in which we are given $N$ observed samples $(\mathbf{x}_i, t_i, s_i, y_i)_{i=1}^N$, each containing $N$ pre-treatment covariates $\{\mathbf{x}_i \in \mathbb{R}^p\}_{i=1}^N$. The treatment variable $t_i$ in this work has various support, e.g., $\{0, 1\}$ for binary treatment settings, $\mathbb{R}$ for continuous treatment settings, and graphs/words for structured treatment settings. For each sample, the potential outcome ($\mu$-model) $\mu(\mathbf{x}, t)$ or $\mu(\mathbf{x}, t, s)$ is the response of the $i$-th sample to a treatment $t$, where in some cases each treatment will be associated with a dosage $s_{t_i} \in \mathbb{R}$. The propensity score ($\pi$-model) is the conditional probability of treatment assignment given the observed covariates $\pi(T = t | X = \mathbf{x})$. The above two models can be parameterized as $\mu_\theta$ and $\pi_\phi$, respectively. The task is to estimate the Average Dose Response Function (ADRF): $\mu(\mathbf{x}, t) = \mathbb{E}[Y | X = \mathbf{x}, do(T = t)]$ (Shoichet, 2006), which includes special cases in discrete treatment scenarios that can also be estimated as the average treatment effect (ATE): $ATE = \mathbb{E}[\mu(\mathbf{x}, 1) - \mu(\mathbf{x}, 0)]$ and its individual version ITE.

What makes the above problem more challenging than supervised learning is that we never see the missing counterfactuals and ground truth causal effects in observational data. Therefore, we first introduce the required fundamentally important assumptions that give the strongly ignorable condition such that statistical estimands can be interpreted causally.

**Assumption 3.1.** (Ignorability/Unconfoundedness) implies no hidden confounders such that $Y(T = t) \perp\!\!\!\perp T | X$. In the binary treatment case, $Y(0), Y(1) \perp\!\!\!\perp T | X$.

**Assumption 3.2.** (Positivity/Overlap) The treatment assignment is non-deterministic such that, i.e. $0 < \pi(t|x) < 1, \forall x \in \mathcal{X}, t \in \mathcal{T}$

Assumption 3.1 ensures the causal effect is identifiable, implying that treatment is assigned independent of the potential outcome and randomly for every subject regardless of its covariates, which allows estimating ADRF using $\mu(t) := \mathbb{E}[Y | do(T = t)] = \mathbb{E}[\mathbb{E}[[Y | \mathbf{x}, T = t]]$ (Rubin, 1978). One naive

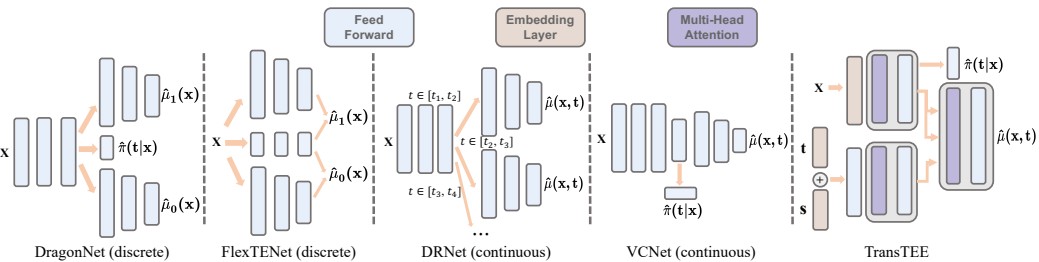

Figure 2: **A schematic comparison** of TransTEE and recent works including DragonNet(Shi et al., 2019), FlexTENet(Curth & van der Schaar, 2021b), DRNet(Schwab et al., 2020) and VCNet(Nie et al., 2021). TransTEE handles all the scenarios without handcrafting treatment-specific architectures and any additional parameter overhead.

estimator of $\mu(\mathbf{x}, t) = \mathbb{E}[Y|X = \mathbf{x}, T = t]$ is the sample average $\mu(t) = \sum_{i=1}^{n} \hat{\mu}(\mathbf{x}_i, t)$. Assumption 3.2 states that there is a chance of seeing units in every treated group.

# 4 TRANSTEE: TRANSFORMERS AS TREATMENT EFFECT ESTIMATORS

The systematic similarity of potential outcomes of different treatment groups is important for TEE (Curth & van der Schaar, 2021b). Note that $\mathbf{x}$ is often high-dimensional while $t$ is not, which means naively feeding $(\mathbf{x}, t)$ to MLPs is not favorable since the impacts of treatment tend to be lost. As a result, various architectures and regularizations have been proposed to enforce structural similarity and differences among treatment groups. However, they are limited to specific use cases as shown in Section 2 and Figure 2. To remedy it, we use three simple yet effective design choices based on attention mechanisms. The resulting scalable framework TransTEE can tackle the problems of most existing treatment effect estimators (e.g., multiple/continuous/structured treatments, treatments interaction, and treatments with dosage) without ad-hoc architectural designs, e.g., multiple branches.

**Preliminary.** The main module in TransTEE is the attention layer Vaswani et al. (2017): given $d$-dimensional query, key, and value matrices $Q \in \mathbb{R}^{d \times d_k}, K \in \mathbb{R}^{d \times d_k}, V \in \mathbb{R}^{d \times d_v}$, attention mechanism computes the outputs as $\mathcal{H}(Q, K, V) = \text{softmax}(\frac{QK^T}{\sqrt{d_k}})V$. In practice, multi-head attention is preferable to jointly attend to the information from different representation subspaces.

$$\mathcal{H}_M(Q, K, V) = \text{Concat}(head_1, ..., head_h)W^O, \text{where } head_i = \mathcal{H}(QW_i^Q, KW_i^K, VW_i^V),$$

where $W_i^Q \in \mathbb{R}^{d \times d_k}, W_i^V \in \mathbb{R}^{d \times d_k}, W_i^V \in \mathbb{R}^{d \times d_v}$ and $W^O \in \mathbb{R}^{hd_v \times d}$ are learnable matrices.

## 4.1 COVARIATE AND TREATMENT EMBEDDING LAYERS

**Treatment Embedding Layer.** As illustrated in Figure 2 and Table. 1, as treatments are often of much lower dimension compared to covariates, to avoid missing the impacts of treatments, previous works (e.g., DragonNet (Shi et al., 2019), FlexTENet (Curth & van der Schaar, 2021b), DRNet (Schwab et al., 2020)) assign covariates from different treatment groups to different branches, which is *highly parameter inefficient*. Besides, We analyze in Proposition 2 (Appendix D) that, for continuous treatments/dosages, the performance is affected by both number of branches and the value interval of treatment. However, almost all previous works on continuous treatment/dosage assume the treatment or dosage is in a fixed value interval e.g., $[0, 1]$ and Figure 3 shows that prevalent works *fail when tested under shifts of treatments*. These two observations motivate us to use two learnable linear layers to project scalar treatments and dosages to $d$-dimension vectors separately: $M_t = \text{Linear}(t), M_s = \text{Linear}(s)$, where $M_t \in \mathbb{R}^d$. $M_s \in \mathbb{R}^{\tilde{d}}$ exists just when each treatment has a dosage parameter, otherwise, only treatment embedding is needed. When multiple ($n$) treatments act simultaneously, the projected matrix will be $M_t \in \mathbb{R}^{\tilde{d} \times n}, M_s \in \mathbb{R}^{d \times n}$ and when facing structural treatments (languages, graphs), the treatment embedding will be projected by language models and graph neural networks respectively. By using the treatment embeddings, TransTEE is shown to be (i) *robust under treatment shifts*, and (ii) *parameter-efficient*.

**Covariates Embedding Layer.** Different from previous works that embed all covariates by one fully connected layer, where the differences between covariates tend to be lost, and is hard to study

the function of an individual covariate in a sample. TransTEE learns different embeddings for each covariate, namely $M_x = \text{Linear}(\mathbf{x})$, and $M_x \in \mathbb{R}^{d \times p}$, where $p$ is the number of covariate. Covariates embedding enables us to study the effect of individual covariate on the outcome.

## 4.2 COVARIATE AND TREATMENT SELF-ATTENTION

For covariates, prevalent methods represent covariates as a whole feature using MLPs, where pairwise covariate interactions are lost when adjusting covariates. Therefore, we cannot study the effect of each covariate on the estimated result. In contrast, TransTEE processes each covariate embedding independently and model their interactions by self-attention layers. Namely,

$$\hat{M}_x^l = \mathcal{H}_M(M_x^{l-1}, M_x^{l-1}, M_x^{l-1}) + M_x^{l-1}, M_x^l = \text{MLP}(\text{BN}(\hat{M}_x^l)) + \hat{M}_x^l.$$

where $M_x^l$ is the output of $l$ layer and BN is the BatchNorm layer. Simultaneously, the treatments and dosages embeddings are concatenated and projected to the latent dimension by a linear layer, which generates a new embedding $M_{st} \in \mathbb{R}^d$. Then self-attention is applied

$$M_{st}^l = \mathcal{H}_M(M_{st}^{l-1}, M_{st}^{l-1}, M_{st}^{l-1}) + M_{st}^{l-1}, M_{st}^l = \text{MLP}(\text{BN}(\hat{M}_{st}^l)) + \hat{M}_{st}^l.$$

The self-attention layer for treatments enables treatment interactions, an important desideratum for S- and T-learners. Namely, TransTEE can *model the scenario where multiple treatments are applied and attain strong practical utility*, e.g., multiple prescriptions in healthcare or different financial measures in economics. This is an effective remedy for existing methods which are limited to settings where various treatments are not used simultaneously.

## 4.3 TREATMENT-COVARIATE CROSS-ATTENTION

One of the fundamental challenges of causal meta-learners is to model treatment-covariate interactions. TransTEE realizes this by a cross-attention module, treating $M_{st}$ as query and $M_x$ as key and value

$$\hat{M}^l = \mathcal{H}_M(M_{st}^{l-1}, M_x^{l-1}, M_x^{l-1}) + M^{l-1},$$
$$M^l = \text{MLP}(\hat{M}^l) + \hat{M}^l, \hat{y} = \text{MLP}(\text{Pooling}(M^L)),$$

where $M^L$ is the output of the last cross-attention layer and $M^0 = M_{st}^L$. The above interactions are particularly important for adjusting proper covariate or confounder sets for estimating treatment effects (VanderWeele, 2019), which empirically yields *suitable covariate adjustment principles (the Disjunctive Cause Criteria) (De Luna et al., 2011; VanderWeele, 2019) about pre-treatment covariates and confounders* as intuitively illustrated in Figure 1 and corroborated in our experiments.

Denote $\hat{y} := \mu_\theta(\mathbf{x}, t)$ and the training objective is the mean square error (MSE) of the outcome regression: $\mathcal{L}_\theta(\mathbf{x}, y, t) = \sum_{i=1}^n (y_i - \mu_\theta(\mathbf{x}_i, t_i))^2$.

**Remark.** We include an illustration of TransTEE by a concrete example in Appendix B. Note that, although the embedding technique and attention mechanisms are commonly used in Computer Vision, Neural Language Processing communities, it is not well understood *how to guide the design of these modules for causal inference* and *why these techniques benefit TEE tasks are underexplored*. In this work, through the flexible use of embedding and attention mechanisms we design a strong TEE architecture, we further use conceptual analysis and empirical results to show the benefit brought by the used design choices. Besides, when combined with the strong modeling capacity of Transformers, *TransTEE can be extended to high-dimensional data flexibly and effectively* on structured data. The generalizability of the TransTEE also allows new applications like auditing language models beyond semi-synthetic settings as shown in the next section.

## 5 EXPERIMENTAL RESULTS

We elaborate on basic experimental settings, results, analysis, and empirical studies in this section. See Appendix E for full details of all experimental settings and detailed definitions of metrics. See Appendix F for many more results and remarks.

### 5.1 EXPERIMENTAL SETTINGS

**Datasets.** Since the true counterfactual outcome (or ADRF) are rarely available for real-world data, we use synthetic or semi-synthetic data for empirical evaluation. for continuous treatments,

Table 2: **Experimental results comparing NN-based methods on the IHDP datasets,** where ——— means the model is not suitable for continuous treatments. We report the results based on 100 repeats, and numbers after $\pm$ are the estimated standard deviation of the average value. For the vanilla setting with binary treatment, we report the mean absolute difference between the estimated and true ATE. For Extrapolation ($h = 2$), models are trained with $t \in [0.1, 2.0]$ and tested in $t \in [0, 2.0]$. For Extrapolation ($h = 5$), models are trained with $t \in [0.25, 5.0]$ and tested in $t \in [0, 5]$.

| METHODS | VANILLA (BINARY) | VANILLA ($h = 1$) | EXTRAPOLATION ($h = 2$) | VANILLA ($h = 5$) | EXTRAPOLATION ($h = 5$) |
|---|---|---|---|---|---|
| TARNET | $0.3670 \pm 0.61112$ | $2.0152 \pm 1.07449$ | $12.967 \pm 1.78108$ | $5.6752 \pm 0.53161$ | $31.523 \pm 1.5013$ |
| DRNET | $0.3543 \pm 0.60622$ | $2.1549 \pm 1.04483$ | $11.071 \pm 0.99384$ | $3.2779 \pm 0.42797$ | $31.524 \pm 1.50264$ |
| FLEXTENET | $0.2700 \pm 0.10000$ | ——— | ——— | ——— | ——— |
| VCNET | $0.2098 \pm 0.18236$ | $0.7800 \pm 0.61483$ | NAN | NAN | NAN |
| TRANSTEE | $\mathbf{0.0983 \pm 0.15384}$ | $0.1151 \pm 0.10289$ | $0.2745 \pm 0.14976$ | $0.1621 \pm 0.14443$ | $0.2066 \pm 0.23258$ |
| TRANSTEE+MLE | $0.1721 \pm 0.40061$ | $0.0877 \pm 0.03352$ | $0.2685 \pm 0.17552$ | $0.2079 \pm 0.17637$ | $0.1476 \pm 0.07123$ |
| TRANSTEE+TR | $0.1913 \pm 0.29953$ | $0.0781 \pm 0.03243$ | $0.2393 \pm 0.08154$ | $\mathbf{0.1143 \pm 0.03224}$ | $\mathbf{0.0947 \pm 0.0824}$ |
| TRANSTEE+PTR | $0.2193 \pm 0.34667$ | $\mathbf{0.0762 \pm 0.07915}$ | $\mathbf{0.2352 \pm 0.17095}$ | $0.1363 \pm 0.08036$ | $0.1363 \pm 0.08035$ |

we use one synthetic dataset and two semi-synthetic datasets: the *IHDP* and *News* datasets. For treatment with continuous dosages, we obtain covariates from a real dataset TCGA (Chang et al., 2013) and generate treatments, where each treatment is accompanied by a dosage. The resulting dataset is named *TCGA (D)*. Following (Kaddour et al., 2021), datasets for structured treatments include *Small-World (SW)*, which contains $1,000$ uniformly sampled covariates and 200 randomly generated Watts–Strogatz small-world graphs (Watts & Strogatz, 1998) as treatments, and *TCGA (S)*, which uses $9,659$ gene expression of cancer patients (Chang et al., 2013) for covariates and $10,000$ molecules from the QM9 dataset (Ramakrishnan et al., 2014) as treatments. For the study on language models, we use the *Enriched Equity Evaluation Corpus (EEEC)* (Feder et al., 2021).

**Baselines.** Baselines for **continuous and binary** treatments include TARnet (Shalit et al., 2017), Dragonnet (Shi et al., 2019), DRNet (Schwab et al., 2020), FlexTENet (Curth & van der Schaar, 2021b), and VCNet (Nie et al., 2021). SCIGAN (Bica et al., 2020b) is chosen as the baseline for **continuous dosages**. Besides, we revise DRNet (Schwab et al., 2020), TARNet (Shalit et al., 2017), and VCNet (Nie et al., 2021) to DRNet (D), TARNet (D), VCNet (D), respectively, which enable multiple treatments and dosages. Specifically, DRNet (D) has $T$ main flows, each corresponding to a treatment and is divided into $B_D$ branches for continuous dosage. Baselines for **structured** treatments include Zero (Kaddour et al., 2021), GNN (Kaddour et al., 2021), GraphITE (Harada & Kashima, 2021), and SIN (Kaddour et al., 2021). To compare the performance of different frameworks fairly, all of the models regress on the outcome with empirical samples without any regularization. For MLE training of the propensity score model, the objective is the negative log-likelihood: $\mathcal{L}_\phi := -\frac{1}{n} \sum_{i=1}^{n} \log \pi_\phi(t_i | \mathbf{x}_i)$.

**Evaluation Metric.** For **continuous and binary** treatments, we use the average mean squared error on the test set. For **structured** treatments, following (Kaddour et al., 2021), we rank all treatments by their propensity $\pi(t|\mathbf{x})$ in a descending order. Top $K$ treatments are selected and the treatment effect of each treatment pair is evaluated by unweighted/weighted expected Precision in Estimation of Heterogeneous Effect (PEHE) (Kaddour et al., 2021), where the WPEHE@K accounts for the fact that treatment pairs that are less likely to have higher estimation errors should be given less importance. For **multiple treatments and dosages**, AMSE is calculated over all dosage and treatment pairs, resulting in $\text{AMSE}_\mathcal{D}$.

## 5.2 CASE STUDY AND NUMERICAL RESULTS

**Case study on treatment distribution shifts** We start by conducting a case study on treatment distribution shifts (Figure 3), and exploring an extrapolation setting in which the treatment may subsequently be administered at values never seen before during training. Surprisingly, we find that while standard results rely on constraining the values of treatments Nie et al. (2021) and dosages Schwab et al. (2020) to a specific range, our methods perform surprisingly well when extrapolating beyond these ranges as assessed on several benchmarks. By comparison, other methods appear comparatively brittle in these same settings. See Appendix D for detailed discussion.

**Case study of propensity modeling.** TransTEE is conceptually simple and effective. However, when the sample size is small, it becomes important to account for selection bias (Alaa & Schaar, 2018). However, most existing regularizations can only be used when the treatments

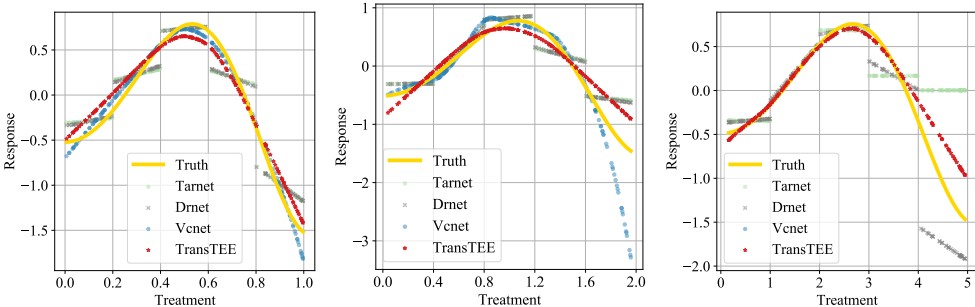

(a) $h = 1$ in training and testing. (b) $h = 1.75$ in training and $h = $ (c) $h = 5$ in training and testing.
2 in testing (extrapolation).

Figure 3: **Estimated ADRF on the synthetic dataset**, where treatments are sampled from an interval $[l, h]$, where $l = 0$.

are discrete (Bica et al., 2020a; Kallus, 2020; Du et al., 2021). Thus we propose two regularization variants for continuous treatment/dosages, which are termed Treatment Regularization (TR, $\mathcal{L}_\phi^{TR}(\mathbf{x}, t) = \sum_{i=1}^n \left( t_i - \pi_\phi(\hat{t}_i|\mathbf{x}_i) \right)^2$) and its probabilistic version Probabilistic Treatment Regularization (PTR, $\mathcal{L}_\phi^{PTR} = \sum_{i=1}^n \left[ \frac{(t_i - \pi_\phi(\mu|\mathbf{x}_i))^2}{2\pi_\phi(\sigma^2|\mathbf{x}_i)} + \frac{1}{2} \log \pi_\phi(\sigma^2|\mathbf{x}_i) \right]$) respectively. The overall model is trained in a adversarial pattern, namely $\min_\theta \max_\phi \mathcal{L}_\theta(\mathbf{x}, y, t) - \mathcal{L}_\phi(\mathbf{x}, t)$. Specifically, a propensity score model $\pi_\phi(t|\mathbf{x})$ parameterized by an MLP is learned by minimizing $\mathcal{L}_\phi(\mathbf{x}, t)$, and then the outcome estimators $\mu_\theta(\mathbf{x}, t)$ is trained by $\min_\theta \mathcal{L}_\theta(\mathbf{x}, y, t) - \mathcal{L}_\phi(\mathbf{x}, t)$. To overcome selection biases over-representation space, the bilevel optimization enforces effective treatment effect estimation while modeling the discriminative propensity features to partial out parts of covariates that cause the treatment but not the outcome and dispose of nuisance variations of covariates (Kaddour et al., 2021). Such a recipe can account for *selection bias* where $\pi(t|\mathbf{x}) \neq p(t)$ and leave spurious correlations out, which can also be more robust under model misspecification especially in the settings that require extrapolation on treatment (See Table 2 and Appendix C for concrete formalisms and discussions.).

As in Table 2, Appendix Table 5 and Table 12, with the addition of adversarial training as well as TR and PTR, TransTEE's estimation error with continuous treatments can be further reduced. Overall, TR is better in the continuous case with smaller treatment distribution shifts, while PTR is preferable when shifts are greater. Both TR and PTR cannot bring performance gains over discrete cases. The superiority of TR and PTR in combination with TransTEE over comprehensive existing works, especially in semi-synthetic benchmarks like IHDP that may systematically favor some types of algorithms over others (Curth et al., 2021), also calls for more understanding of NNs' inductive biases in treatment effect estimation problems of interest. Moreover, covariate selection visualization in TR and PTR (Figure 4(a) , Table 4 and Appendix F) supports the idea that modeling the propensity score effectively promotes covariate adjustment and partials out the effects from the covariates on the treatment features. We also compare the training dynamic of different regularizations in Appendix F, where TR and PTR are further shown able to improve the convergence of TransTEE.

**Continuous treatments.** To evaluate the efficiency with which TransTEE estimates the average dose-response curve (ADRF), we compare against other recent NN-based methods (Tables 2). Comparing results in each column, we observe performance boosts for TransTEE. Further, TransTEE attains a much smaller error than baselines in cases where the treatment interval is not restricted to $[0, 1]$ (e.g., $t \in [0, 5]$) and when the training and test treatment intervals are different (extrapolation). Interestingly, even vanilla TransTEE produces competitive performance compared with that of $\pi(t|\mathbf{x})$ trained additionally using MLE, demonstrating the ability of TransTEE to effectively model treatments and covariates. The estimated ADRF curves on the IHDP and News datasets are shown in Figure 11 and Figure 13 in Appendix. TARNet and DRNet produce discontinuous ADRF estimators and VCNet only performs well when $t \in [0, 1]$. However, TransTEE attains lower estimation error and preserves the continuity of ADRF on different treatment intervals.

**Continuous dosage.** In Table 5, we compare TransTEE against baselines on the TCGA (D) dataset with default treatment selection bias 2.0 and dosage selection bias 2.0. As the number of treatments

Table 3: **Effect of Gender (top) and Race (bottom) on POMS classification with the EEEC dataset**, where $\text{ATE}_{GT}$ is the ground truth ATE based on 3 repeats with confidence intervals [CI] constructed using standard deviations.

| | | Correlation/Representation Based Baselines | | | Treatment Effect Estimators | | | |
|---|---|---|---|---|---|---|---|---|
| TC | $\text{ATE}_{GT}$ | TReATE | CONEXP | INLP | TarNet | DRNet | VCNet | TransTEE |
| Gender | 0.086 | 0.125 | 0.02 | 0.313 | 0.0067 | 0.0088 | 0.0085 | **0.013** |
| [CI] | [0.082,0.09] | [0.110,0.14] | [0.0,0.05] | [0.304,0.321] | [0.0049,0.0076] | [0.0084,0.009] | [0.0036, 0.0111] | [0.008, 0.0168] |
| Race | 0.014 | 0.046 | 0.08 | 0.591 | 0.005 | 0.006 | 0.003 | **0.0174** |
| [CI] | [0.012,0.016] | [0.038,0.054] | [0.02,0.014] | [0.578,0.605] | [0.0021, 0.0069] | [0.0047, 0.0081] | [0.0025, 0.0037] | [0.0113, 0.0238] |

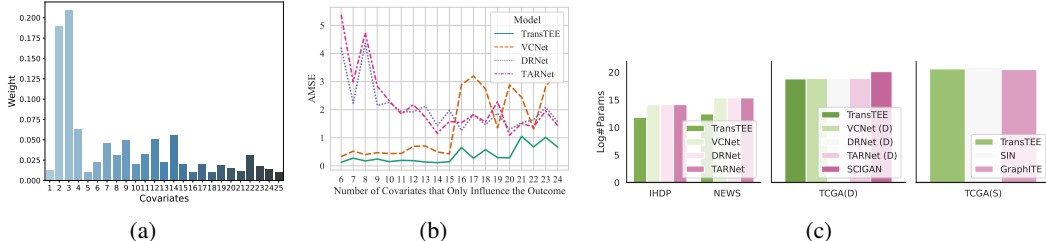

(a)          (b)          (c)

Figure 4: (a) The learned weights of the cross-attention module on IHDP dataset. TransTEE adjusts confounders $S_{con} = \{1, 2, 3, 5, 6\}$ properly with higher weights during the cross attention process. (b) AMSE attained by models on IHDP with different numbers of noisy covariates. (c) Number of parameters for different models on four different datasets, where the log on the y-axis is base 2.

increases, TransTEE and its variants (with regularization term) consistently outperform the baselines by a large margin on both training and test data. TransTEE's effectiveness is also shown in Appendix Figure 8, where the estimated ADRF curve of each treatment considering continuous dosages is plotted. Compared to baselines, TransTEE attains better results over all treatments. Stronger selection bias in the observed data makes estimation more difficult because it becomes less likely to see certain treatments or particular covariates. Considering different dosages and treatment selection bias, Appendix Figure 7 shows that as biases increase, TransTEE consistently performs the best.

**Structured treatments.** We compared the performance of TransTEE to baselines on the training and test set of both SW and TCGA datasets with varying degrees of treatment selection bias. The numerical results are shown in Appendix Table 13. The performance gain between GNN and Zero indicates that taking into account graph information significantly improves estimation. The results suggest that, overall, the performance of TransTEE is the best due to the strong modeling capability and advanced model structure for processing high-dimensional treatments. SIN is the best model among these baselines. However, when the bias is equal to $0.1$, SIN fails to attain estimation results better than the Zero baseline. To evaluate each model's robustness to varying levels of selection bias, performance curve with $\kappa \in [0, 40]$ for the SW dataset and $\kappa \in [0, 0.5]$ for the TCGA dataset are shown in Figure 14 and Figure 15 in Appendix. Considering both metrics, TransTEE outperforms baselines by a large margin across the entire range of evaluated treatment selection biases.

### 5.3 ANALYSIS

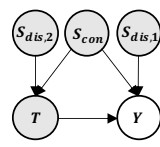

Figure 5: The causal graph of IHDP dataset.

**Analysis of covariate adjustment of cross-attention module.** TransTEE embeds each covariate independently and then make treatments select proper covariates for prediction by cross-attention. The resulting interpretability of the covariate adjustment process using attention weights is one clear advantage over existing works. Thus we visualize the covariate selection results (cross-attention weights) in Figure 4(a). As elaborated in Appendix E.3, the IHDP dataset has 25 covariates, which is divided into 3 groups: $S_{con} = \{1, 2, 3, 5, 6\}$, $S_{dis,1} = \{4, 7 \sim 15\}$, and $S_{dis,2} = \{16 \sim 25\}$. $S_{con}$ influences both $T$ and $Y$, $S_{dis,1}$ influences only $Y$, and $S_{dis,1}$ influences only $T$. Covariates in $S_{dis,1}$ are named noisy covariates since they have no correlation with the treatment. Their causal relationships are illustrated in Figure 5. Interestingly, confounders $S_{con}$ are assigned higher weights while noisy covariates (those influence the outcome but are irrelevant to the treatment) lower $S_{dis,1}$, which matches the principles in (VanderWeele, 2019) and corroborate the ability of TransTEE to estimate treatment effects in

complex datasets by controlling both pre-treatment variables and confounders properly. Moreover, Figure 4(b) shows that TransTEE consistently outperforms baselines across different numbers of noisy covariates.

We further conduct 10 repetitions for TransTEE and its TR and PTR counterparts as reported in Table 4 (Appendix Figure 10 visualizes their cross-attention weights). Denote $w_{con}, w_1, w_2$ as the summation of weights assigned to $S_{con}, S_{dis,1}, S_{dis,2}$ respectively. We can see that, incorporated with both TR and PTR regularization, TransTEE assigns more weights to confounding covariates ($S_{con}$) and fewer weights on noisy covariates, which further verifies the compatibility of TransTEE with propensity score modeling since both TR and PTR improve confounding

Table 4: **Attention weights** for $S_{con}$, $S_{dis,1}$, and $S_{dis,2}$ respectively.

|  | $w_{con}$ | $w_1$ | $w_2$ |
|---|---|---|---|
| **TransTEE** | 0.27 | 0.37 | 0.36 |
| **+TR** | **0.59** | **0.20** | 0.21 |
| **+PTR** | 0.32 | 0.33 | 0.35 |

control. Moreover, TR is better than PTR since it also reduces $w_2$ by a larger margin. This observation gives a suggestion that we should systematically probe TR and PTR besides comparing their numerical performance, especially in settings where the unconfoundedness assumption is violated (Ding et al., 2017) and controlling instrumental variables will incur biases in TEE.

**Amount of model parameters comparison.** The experiment is to corroborate the conceptual comparison in Table 1. We find that the proposed TransTEE has consistently fewer parameters than baselines on all the settings as shown in Figure 4(c). Besides, increasing the number of treatments allows more accurate approximation for continuous treatments/dosages, most of these baselines need to increase branches which incurs parameter redundancy. However, TransTEE is much more efficient.

### 5.4 EMPIRICAL STUDY ON PRE-TRAINED LANGUAGE MODELS

To evaluate the real-world utility of TransTEE, in this subsection, we demonstrate an initial attempt for auditing and debugging large pre-trained language models, an important use case in NLP that is beyond semi-synthetic settings and under-explored in the causal inference literature. Specifically, we use TransTEE to estimate the treatment effects for detecting the effects of domain-specific factors of variation (such as the change of subject's attributes in a sentence) on the predictions of pre-trained language models. We experiment with BERT (Kenton & Toutanova, 2019) (e.g., racial and gender-related nouns) over natural language on the (real) EEEC dataset. We use both the correlation/representation-based baselines introduced in (Feder et al., 2021) and implement treatment effect estimators (e.g., TARnet, DRNet, VCNet, and the proposed TransTEE).

Interestingly, results in Table 3 show that TransTEE effectively estimates the treatment effects of domain-specific variation perturbations even without substantive downstream fine-tuning on specialized datasets. TransTEE outperforms baselines adapted from MLP. Moreover, we showcase the top-k samples with the maximal/minimal ITE and analysis in Appendix F.3. The results show that TransTEE has the potential to provide estimators for practical use cases in predicting model predictions (Ilyas et al., 2022). For example, those identified samples can provide actionable insights like function as contrast sets for analyzing and understanding LMs (Gardner et al., 2020; Abraham et al., 2022) and TransTEE can estimate ATE to enforce invariant or fairness constraints for LMs (Veitch et al., 2021) in a lightweight and efficient manner, which we leave for future work.

### 6 CONCLUDING REMARKS

In this work, we show attention mechanisms can be effective and versatile design choices for TEE tasks. Extensive experiments well verify the effectiveness and utility of the proposed TransTEE, which also imply that more challenging and unified evaluation alternatives of TEE are needed. Moreover, we hope that our findings can lay the groundwork for future work in developing advanced machine learning techniques like pre-training in large-scale observational data in estimating treatment effects, where TransTEE can serve as an effective backbone. Similar to almost all the causal inference methods on observational data, one potential limitation of TransTEE is the reliance on the ignorability assumption. Therefore, one important future direction is extending TransTEE to settings with more complex causal graphs and generate identifiable causal functionals tractable for optimization (Jung et al., 2020) supported by identification theory. Since adjusting covariates without accounting for the causal graph might yield inaccurate or biased estimates of the causal effect (Pearl, 2009), how to integrate TransTEE with domain knowledge (Imbens & Rubin, 2015) for alleviating its potential negative societal impacts in consequential decision making will also be important.

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

# Exploring Transformer Backbones for Heterogeneous Treatment Effect Estimation

CONTENTS

## A    EXTENDED RELATED WORK

**Propensity Score.** Most related works fundamentally rely on strongly ignorable conditions. Still even under ignorability, treatments may be selectively assigned according to propensities that depend on the covariates. To overcome the impact of such confounding, many statistical methods (Austin, 2011) like covariate adjustment (Austin, 2011), matching (Rubin & Thomas, 1996; Abadie & Imbens, 2016), stratification (Frangakis & Rubin, 2002), reweighting (Hirano et al., 2003), g-computation (Imbens & Rubin, 2015), have been proposed. More recent approaches include propensity dropout (Alaa et al., 2017), and multi-task Gaussian process (Alaa & van der Schaar, 2017). Explicitly modeling the propensity score, which reflects the underlying policy for assigning treatments to subjects, has also shown to be effective in reasoning about the unobserved counterfactual outcomes and accounting for confounding. Based upon it, double robust estimators and targeted regularization are proposed to guarantee the consistency of estimated treatment effects under misspecification of either the outcome or propensity score model (Kang & Schafer, 2007; Funk et al., 2011). There are also works using adversarial training for balanced representations (Bica et al., 2020a; Kallus, 2020; Du et al., 2021). However, most traditional approaches are restricted to binary treatments and the capacity of NNs for such problems have not been fully leveraged.

**Domain Adaptation** There are some close connections between causal inference and domain adaptation, in particular, out-of-distribution robustness. Intuitively, traditional domain adversarial training learns representations that are indistinguishable by the domain classifier by minimizing the worst-domain empirical error (Ganin et al., 2016; Zhao et al., 2018; Wang et al., 2022; Zhang et al., 2022). The algorithmic insights can be handily translated to the TEE domain (Shalit et al., 2017; Johansson et al., 2020; Feder et al., 2021). Here we also have the desideratum that covariate representations should be balanced such that the selection bias is minimized and the effect is maximally determined by the treatment. Algorithmically, when the treatment is continuous, we connect our method to continuously indexed domain adaptation (Wang et al., 2020). Our formulation and algorithm also serve to build connections to a diverse set of statistical thinking on causal inference and domain adaptation, of which much can be gained by mutual exchange of ideas (Johansson et al., 2020). Explicitly modeling the propensity score also seeks to connect causal inference with transfer learning to inspire domain adaptation methodology and holds the potential to handle a wider range of problems like hidden stratification in domain generalization, which we leave for future work.

**Comparision between TransTEE and ANU (Xu et al., 2022).** (i) The model structure is different. ANU performs cross-attention between $z_x$, and $z_t$, and no self-attention is applied. However, TransTEE performs self-attention on $z_x, z_t$ respectively and then cross-attention is performed between $z_x, z_t$. When facing high-dimensional data, such as texts, images, and graphs, without multiple self-attention layers on $z_x, z_t$ separately, the representations will be weak. That is why in machine translation, object detection, and segmentation tasks, the representations of images/texts will be firstly processed by multiple self-attention layers and then perform cross-attention with queries. We will verify this point in the following experiments. (ii) ANU cannot be applied to multi-treatment settings, which have been extensively studied recently (Kaddour et al., 2021; Bica et al., 2020b; Parbhoo et al., 2021). The comparison experiments are in Section F.1.

## B    AN ILLUSTRATIVE EXAMPLE

To better understand the workflow with the above designs, we present a simple illustration here. Consider a use case in medicine effect estimation, where **x** contains $p$ patient information, e.g., *Age, Sex, Blood Pressure (BP), and Previous infection condition (Prev)* with a corresponding causal graph (Figure 1). $n$ medicines (treatments) are applied simultaneously and each medicine has a corresponding dosage. As shown in Figure 6, each covariate, treatment, and dosage will first be embedded to $d$-dimension representation by a specific **learnable embedding layer**. Each treatment embedding will be concatenated with its dosage embedding and the concatenated feature will be projected by a linear layer to produce $d$ dimensional vectors. **Self-**

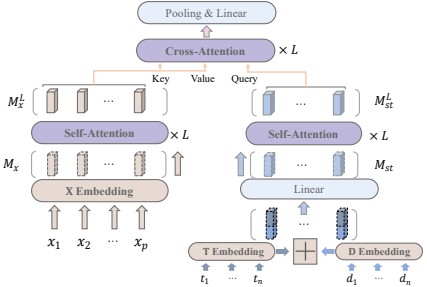

Figure 6: An Illustrative Example about the workflow of TransTEE.

**attention modules** optimizes these embeddings by aggregating contextual information. Specifically, attribute *Prev* is more related to *age* than *sex*, hence the attention weight of *Prev* feature to *age* feature is larger and the update of *Prev* feature will be more dependent on the *age* feature. Similarly, the interaction of multi-medicines is also attained by the self-attention module. The last **Cross-attention module** enables treatment-covariate interactions, which is shown in Figure 2 that, each medicine will assign a higher weight to relevant covariates especially confounders (*BP*) than irrelevant ones. Finally, we pool the resulted embedding and use one linear layer to predict the outcome.

## C    DETAILS AND DISCUSSIONS ABOUT PROPENSITY SCORE MODELLING

We first discuss the fundamental differences and common goals between our algorithm and traditional ones: as a general approach to causal inference, TransTEE can be directly harnessed with traditional methods that estimate propensity scores by including hand-crafted features of covariates (Imbens & Rubin, 2015) to reduce biases through covariate adjustment (Austin, 2011), matching (Rubin & Thomas, 1996; Abadie & Imbens, 2016), stratification (Frangakis & Rubin, 2002), reweighting (Hirano et al., 2003), g-computation (Imbens & Rubin, 2015), sub-classification (Rosenbaum & Rubin, 1984), covariate adjustment (Austin, 2011), targeted regularization (Van Der Laan & Rubin, 2006) or conditional density estimation (Nie et al., 2021) that create quasi-randomized experiments (D'Agostino, 1998). It is because the general framework provides an advantage to using an off-the-shelf propensity score regularizer for balancing covariate representations. Similar to the goal of traditional methods like inverse probability weighting and propensity score matching (Austin, 2011), which seeks to weigh a single observation to mimic the randomization effects with respect to the covariate from different treatment groups of interest.

Unlike previous works that use hand-crafted features or directly model the conditional density via maximum likelihood training, which is prone to high variance when handling high-dimensional, structured treatments (Singh et al., 2019) and can be problematic when we want to estimate a plausible propensity score from the generative model (Mohamed & Lakshminarayanan, 2016) (see the degraded performance of MLE in Table 2), TransTEE learns a propensity score network $\pi_\phi(t|\mathbf{x})$ via minimax bilevel optimization. The motivations for adversarial training between $\mu_\theta(\mathbf{x}, t)$ and $\pi_\phi(t|\mathbf{x})$ are threefold: (i) it enforces the independence between treatment and covariate representations as shown in Proposition 1, which serves as algorithmic randomization in replace of costly randomized controlled trials (Rubin, 2007) for overcoming selection bias (D'Agostino, 1998; Imbens & Rubin, 2015); (ii) it explicitly models propensity $\pi_\phi(t|\mathbf{x})$ to refine treatment representations and promote covariate adjustment (Kaddour et al., 2021); and (iii) taking an adversarial domain adaptation perspective, the methodology is effective for learning invariant representations and further regularizes $\mu_\theta(\mathbf{x}, t)$ to be invariant to nuisance factors and may perform better empirically on some classes of distribution shifts (Ganin et al., 2016; Shalit et al., 2017; Zhao et al., 2018; Johansson et al., 2020; Wang et al., 2020).

Based on the above discussion, when treatments are discrete, one might consider directly applying heuristic methods like adversarial domain adaptation (see (Ganin et al., 2016; Zhao et al., 2018) for algorithmic development guidelines). We note the heuristic nature of domain-adversarial methods (see (Wu et al., 2019) for clear failure cases), and a debunking of the common claim that (Ben-David et al., 2010) guarantees the robustness of such methods. Here, we focus on continuous TEE, a more general and challenging scenario, where we want to estimate ADRF, and propose two variants of $\mathcal{L}_\phi$ as an adversary for the outcome regression objective $\mathcal{L}_\theta$ accordingly. Recall that $\mathcal{L}_\theta(\mathbf{x}, y, t) = \sum_{i=1}^{n} (y_i - \mu_\theta(\mathbf{x}_i, t_i))^2$, the adversarial training process is shown in Eq. 1 below:

$$\min_\theta \max_\phi \mathcal{L}_\theta(\mathbf{x}, y, t) - \mathcal{L}_\phi(\mathbf{x}, t). \tag{1}$$

We refer to the above minimax game for algorithmic randomization in replace of costly randomized controlled trials. Such algorithmic randomization based on neural representations using propensity score creates subgroups of different treated units as if they had been randomly assigned to different treatments such that conditional independence $T \perp\!\!\!\perp X \mid \pi(T|X)$ is enforced across strata and continuation, which approximates a random block experiment to the observed covariates (Imbens & Rubin, 2015).

Below we introduce two variants of $\mathcal{L}_\phi(\mathbf{x}, t)$:

Table 5: **Performance of individualized treatment-dose response estimation** on the TCGA (D) dataset with different numbers of treatments. We report AMSE and standard deviation over 30 repeats. The selection bias on treatment and dosage are both set to be 2.0.

| METHODS | #TREATMENT=1 | | #TREATMENT=2 | | #TREATMENT=3 | |
|---|---|---|---|---|---|---|
| | IN-SAMPLE | OUT-SAMPLE | IN-SAMPLE | OUT-SAMPLE | IN-SAMPLE | OUT-SAMPLE |
| SCIGAN | $5.6966 \pm 0.0000$ | $5.6546 \pm 0.0000$ | $2.0924 \pm 0.0000$ | $2.3067 \pm 0.0000$ | $4.3183 \pm 0.0000$ | $4.6231 \pm 0.0000$ |
| TARNET(D) | $0.7888 \pm 0.0609$ | $0.7908 \pm 0.0606$ | $1.4207 \pm 0.0784$ | $1.4206 \pm 0.0777$ | $3.1982 \pm 0.5847$ | $3.1920 \pm 0.5746$ |
| DRNET(D) | $0.8034 \pm 0.0469$ | $0.8052 \pm 0.0466$ | $1.3739 \pm 0.0858$ | $1.3738 \pm 0.0853$ | $2.8632 \pm 0.4227$ | $2.8558 \pm 0.4143$ |
| VCNET(D) | $0.1566 \pm 0.0303$ | $0.1579 \pm 0.0301$ | $0.2919 \pm 0.0743$ | $0.2918 \pm 0.0737$ | $0.6459 \pm 0.1387$ | $0.6493 \pm 0.1397$ |
| TRANSTEE | $0.0573 \pm 0.0361$ | $0.0585 \pm 0.0358$ | $\mathbf{0.0550 \pm 0.0137}$ | $\mathbf{0.0556 \pm 0.0129}$ | $0.2803 \pm 0.0658$ | $0.2768 \pm 0.0639$ |
| TRANSTEE + TR | $0.0495 \pm 0.0176$ | $0.0509 \pm 0.0180$ | $0.0663 \pm 0.0268$ | $0.0671 \pm 0.0268$ | $\mathbf{0.2618 \pm 0.0737}$ | $\mathbf{0.2577 \pm 0.0726}$ |
| TRANSTEE + PTR | $\mathbf{0.0343 \pm 0.0096}$ | $\mathbf{0.0355 \pm 0.0094}$ | $0.0679 \pm 0.0252$ | $0.0686 \pm 0.0252$ | $0.2645 \pm 0.0702$ | $0.2597 \pm 0.0675$ |

**Treatment Regularization (TR)** is a standard MSE over the treatment space given the predicted treatment $\hat{t}_i$ and the ground truth $t_i$

$$\mathcal{L}_\phi^{TR}(\mathbf{x}, t) = \sum_{i=1}^n \left( t_i - \pi_\phi(\hat{t}_i|\mathbf{x}_i) \right)^2. \tag{2}$$

TR is explicitly matching the mean of the propensity score to that of the treatment. In an ideal case, the $\pi(t|\mathbf{x})$ should be uniformly distributed given different $\mathbf{x}$. However, the above treatment regularization procedure only provides matching for the mean of the propensity score, which can be prone to bad equilibriums and treatment misalignment (Wang et al., 2020). Thus, we introduce the distribution of $t$ and model the uncertainty rather than predicting a scalar $t$:

**Probabilistic Treatment Regularization (PTR)** is a probabilistic version of TR which models the mean $\mu$ (with a slight abuse of notation) and variance $\sigma^2$ of estimated treatment $\hat{t}_i$

$$\mathcal{L}_\phi^{PTR} = \sum_{i=1}^n \left[ \frac{(t_i - \pi_\phi(\mu|\mathbf{x}_i))^2}{2\pi_\phi(\sigma^2|\mathbf{x}_i)} + \frac{1}{2} \log \pi_\phi(\sigma^2|\mathbf{x}_i) \right]. \tag{3}$$

The PTR matches the whole distribution, i.e. both the mean and variance, of the propensity score to that of the treatment, which can be preferable in certain cases.

**Equilibrium of the Minimax Game.** We analyze that TR and PTR can align the first and second moment of continuous treatments at equilibrium respectively, and thus promote the independence between treatment $t$ and covariate $\mathbf{x}$. To be clear, we denote $\mu_\theta(\mathbf{x}, t) := w_y \circ (\Phi_x(\mathbf{x}), \Phi_t(t))$ and $\pi_\phi(t|\mathbf{x}) := w_t \circ \Phi_x(\mathbf{x})$, which decompose the predictions into featurizers $\Phi_t : \mathcal{T} \to \mathcal{Z}_T, \Phi_x : \mathcal{X} \to \mathcal{Z}_X$ and predictors $w_y : \mathcal{Z}_X \times \mathcal{Z}_T \to \mathcal{Y}, w_t : \mathcal{Z}_X \to \mathcal{T}$. For example, $\Phi_x(\mathbf{x})$ and $\Phi_t(t)$ can be the linear embedding layer and attention modules in our implementation. The propensity is computed on $\Phi_x(\mathbf{x})$, an intermediate feature representation of $\mathbf{x}$. Similarly, $\mu_\theta(\mathbf{x}, t)$ is computed from $\Phi_t(t)$ and $\Phi_x(\mathbf{x})$. For the ease of our analysis below, we assume the predictors $w_t, w_x$ are fixed.

**Proposition 1.** *(The optimum of propensity score model) In the equilibrium of the game, assuming the outcome prediction model is fixed, then the optimum of TR is achieved when $\mathbb{E}[\Phi_t(t)|\Phi_x(\boldsymbol{x})] = \mathbb{E}[\Phi_t(t)], \forall \Phi_x(\boldsymbol{x})$ via matching the mean of propensity score $\pi(\Phi_t(t)|\Phi_x(\boldsymbol{x}))$ and the marginal distribution $p(\Phi_x(\boldsymbol{x}))$ and the optimum discriminator of PTR is achieved via matching both the mean and variance such that $\mathbb{E}[\Phi_t(t)|\Phi_x(\boldsymbol{x})] = \mathbb{E}[\Phi_t(t)], \mathbb{V}[\Phi_t(t)|\Phi_x(\boldsymbol{x})] = \mathbb{V}[\Phi_t(t)], \forall \Phi_x(\boldsymbol{x})$.*

*Proof.* The proof concerns the analysis of the Equilibrium of the Minimax Game. It is a special case of (Wang et al., 2020) when there are only two players, i.e. $\mu_\theta$ and $\pi_\phi$. We represent treatments explicitly and interpret the connections with combating selection biases. Given the outcome regression model $\mu_\theta$ fixed, the optimal propensity score model $\pi^*$ is

$$\begin{aligned} \pi^* &= \arg\min_\pi \mathcal{L}_\phi(\Phi_x(\mathbf{x}), \Phi_t(t)) \\ &= \arg\min_\pi \mathbb{E}_{(\Phi_x(\mathbf{x}), \Phi_t(t)) \sim p(\Phi_x(\mathbf{x}), \Phi_t(t))} \left( \Phi_t(t) - \pi_\theta \left( \Phi_t(\hat{t})|\mathbf{x} \right) \right)^2 \\ &= \arg\min_\pi \mathbb{E}_{\Phi_x(\mathbf{x}) \sim p(\Phi_x(\mathbf{x}))} \mathbb{E}_{\Phi_t(t) \sim p(\Phi_t(t)|\Phi_x(\mathbf{x}))} \left( \Phi_t(t) - \pi_\theta \left( \Phi_t(\hat{t})|\mathbf{x} \right) \right)^2. \end{aligned} \tag{4}$$

The inner minimum is achieved at $\pi_\theta^* \left(\Phi_t(\hat{t})|\mathbf{x}\right) = \mathbb{E}_{\Phi_t(t)\sim p(\Phi_t(t)|\Phi_x(\mathbf{x}))}[\Phi_t(t)]$ given the following quadratic form:

$$
\mathbb{E}_{(\Phi_x(\mathbf{x}),\Phi_t(t))\sim p(\Phi_x(\mathbf{x}),\Phi_t(t))} \left(\Phi_t(t) - \pi_\theta\left(\Phi_t(\hat{t})|\Phi_\mathbf{x}(\mathbf{x})\right)\right)^2 =
$$
$$
\mathbb{E}_{\Phi_t(t)\sim p(\Phi_t(t)|\Phi_x(\mathbf{x}))}[\Phi_t(t)^2] - 2\pi_\theta\left(\Phi_t(\hat{t})|\mathbf{x}\right)\mathbb{E}_{\Phi_t(t)\sim p(\Phi_t(t)|\Phi_x(\mathbf{x}))}[\Phi_t(t)] + \pi_\theta\left(\Phi_t(\hat{t})|\mathbf{x}\right)^2 . \tag{5}
$$

We assume the above optimum condition of the propensity score model always holds with respect to the outcome regression model during training, then the minimax game in Eq. 1 can be converted to maximizing the inner loop:

$$
\max_\phi -\mathcal{L}_\phi(\mathbf{x}, \Phi_t(t)) = \mathcal{L}_{\phi^*}(\Phi_x(\mathbf{x}), \Phi_t(t))
$$
$$
= \mathbb{E}_{(\Phi_x(\mathbf{x}),\Phi_t(t))\sim p(\Phi_x(\mathbf{x}),\Phi_t(t))}\left(\Phi_t(t) - \mathbb{E}_{\Phi_t(t)\sim p(\Phi_t(t)|\Phi_x(\mathbf{x}))}[\Phi_t(t)]\right)^2
$$
$$
= \mathbb{E}_{\Phi_x(\mathbf{x})\sim p(\Phi_x(\mathbf{x}))}\mathbb{E}_{\Phi_t(t)\sim p(\Phi_t(t)|\Phi_x(\mathbf{x}))\sim p(\Phi_x(\mathbf{x}),\Phi_t(t))}\left(\Phi_t(t) - \mathbb{E}_{\Phi_t(t)\sim p(\Phi_t(t)|\Phi_x(\mathbf{x}))}[\Phi_t(t)]\right)^2
$$
$$
= \mathbb{E}_{\Phi_x(\mathbf{x})\sim p(\Phi_x(\mathbf{x}))}\mathbb{V}_{\Phi_t(t)\sim p(\Phi_t(t)|\Phi_x(\mathbf{x}))}[\Phi_t(t)] = \mathbb{E}_{\Phi_x(\mathbf{x})}\mathbb{V}[\Phi_t(t)|\Phi_x(\mathbf{x})]. \tag{6}
$$

Next we show the difference between Eq. 6 and the variance of the treatment $\mathbb{V}[\Phi_t(t)]$:

$$
\mathbb{E}_{\Phi_x(\mathbf{x})\sim p(\Phi_x(\mathbf{x}))}\mathbb{V}_{\Phi_t(t)\sim p(\Phi_t(t)|\Phi_x(\mathbf{x}))}[\Phi_t(t)] - \mathbb{V}[\Phi_t(t)]
$$
$$
=\mathbb{E}_{\Phi_x(\mathbf{x})\sim p(\Phi_x(\mathbf{x}))}[\mathbb{E}[\Phi_t(t)^2|\Phi_x(\mathbf{x})] - \mathbb{E}[\Phi_t(t)|\Phi_x(\mathbf{x})]^2] - (\mathbb{E}[\Phi_t(t)^2] - \mathbb{E}[\Phi_t(t)]^2)
$$
$$
=\mathbb{E}[\Phi_t(t)]^2 - \mathbb{E}_{\Phi_x(\mathbf{x})}[\mathbb{E}[\Phi_t(t)|\Phi_x(\mathbf{x})]^2] = \mathbb{E}_{\Phi_x(\mathbf{x})}[\mathbb{E}[\Phi_t(t)|\Phi_x(\mathbf{x})]]^2 - \mathbb{E}_{\Phi_x(\mathbf{x})}[\mathbb{E}[\Phi_t(t)|\Phi_x(\mathbf{x})]^2]
$$
$$
\leq\mathbb{E}_{\Phi_x(\mathbf{x})}[\mathbb{E}[\Phi_t(t)|\Phi_x(\mathbf{x})]^2] - \mathbb{E}_{\Phi_x(\mathbf{x})}[\mathbb{E}[\Phi_t(t)|\Phi_x(\mathbf{x})]^2] = 0 \tag{7}
$$

where the last inequality is by Jensen's inequality and the convexity of $\Phi_t(t)^2$. The optimum is achieved when $\mathbb{E}[\Phi_t(t)|\Phi_x(\mathbf{x})]$ is constant w.r.t $\Phi_x(\mathbf{x})$ and so $\mathbb{E}[\Phi_t(t)|\Phi_x(\mathbf{x})] = \mathbb{E}[\Phi_t(t)]$, $\forall\Phi_x(\mathbf{x})$.

The proof process for PTR is similar but includes the derivation of variance matching.

$$
\pi^* = \arg\min_\pi \mathcal{L}_\phi(\Phi_x(\mathbf{x}), \Phi_t(t))
$$
$$
= \arg\min_\pi \mathbb{E}_{(\Phi_x(\mathbf{x}),\Phi_t(t))\sim p(\Phi_x(\mathbf{x}),\Phi_t(t))}\left(\frac{(\mathbb{E}[\Phi_t(t)|\Phi_x(\mathbf{x})] - \Phi_t(t))^2}{2\mathbb{V}[\Phi_t(t)|\Phi_x(\mathbf{x})]} + \frac{\log\mathbb{V}[\Phi_t(t)|\Phi_x(\mathbf{x})]}{2}\right)
$$
$$
= \arg\min_\pi \mathbb{E}_{\Phi_x(\mathbf{x})}\mathbb{E}_{\Phi_t(t)}\left(\frac{(\mathbb{E}[\Phi_t(t)|\Phi_x(\mathbf{x})] - \Phi_t(t))^2}{2\mathbb{V}[\Phi_t(t)|\Phi_x(\mathbf{x})]} + \frac{\log\mathbb{V}[\Phi_t(t)|\Phi_x(\mathbf{x})]}{2}\right), \tag{8}
$$

where $\mathbb{E}_{\Phi_x(\mathbf{x})}$ and $\mathbb{E}_{\Phi_t(t)}$ denote $\mathbb{E}_{\Phi_x(\mathbf{x})\sim p(\Phi_x(\mathbf{x}))}$ and $\mathbb{E}_{\Phi_t(t)\sim p(\Phi_t(t)|\Phi_x(\mathbf{x}))}$ respectively for brevity. The first term can be reduce to a constant given the definition of variance:

$$
\mathbb{E}_{\Phi_x(\mathbf{x})\sim p(\Phi_x(\mathbf{x}))}\mathbb{E}_{\Phi_t(t)\sim p(\Phi_t(t)|\Phi_x(\mathbf{x}))}\left(\frac{(\mathbb{E}[\Phi_t(t)|\mathbf{x}] - \Phi_t(t))^2}{2\mathbb{V}[\Phi_t(t)|\mathbf{x}]}\right)
$$
$$
= \mathbb{E}_{\Phi_x(\mathbf{x})\sim p(\Phi_x(\mathbf{x}))}\left(\frac{\mathbb{V}[\Phi_t(t)|\mathbf{x}]}{2\mathbb{V}[\Phi_t(t)|\mathbf{x}]}\right) = \frac{1}{2}. \tag{9}
$$

The second term can be upper bounded by using Jensen's inequality:

$$
\mathbb{E}_{\Phi_x(\mathbf{x})\sim p(\Phi_x(\mathbf{x}))}\mathbb{E}_{\Phi_t(t)\sim p(\Phi_t(t)|\Phi_x(\mathbf{x}))}\left(\frac{\log\mathbb{V}[\Phi_t(t)|\mathbf{x}]}{2}\right)
$$
$$
\leq \frac{1}{2}\log\left(\mathbb{E}_{\Phi_x(\mathbf{x})\sim p(\Phi_x(\mathbf{x}))}[\mathbb{V}[\Phi_t(t)|\Phi_x(\mathbf{x})]]\right) \tag{10}
$$
$$
\leq \frac{1}{2}\log\left(\mathbb{V}[\Phi_t(t)]\right).
$$

Combining Eq. 9 and Eq. 10, the optimum $\frac{1}{2} + \frac{1}{2}\log\left(\mathbb{V}[\Phi_t(t)]\right)$ is achieved when $\mathbb{E}[\Phi_t(t)|\Phi_x(\mathbf{x})]$, $\mathbb{V}[\Phi_t(t)|\Phi_x(\mathbf{x})]$ is constant w.r.t $\Phi_x(\mathbf{x})$ and so $\mathbb{E}[\Phi_t(t)|\Phi_x(\mathbf{x})] = \mathbb{E}[\Phi_t(t)]$, $\mathbb{V}[\Phi_t(t)|\Phi_x(\mathbf{x})] = \mathbb{V}[\Phi_t(t)]$, $\forall\Phi_x(\mathbf{x})$ according to the equality conditions of the first and second inequality in Eq. 10, respectively. $\qquad\square$

# D  ANALYSIS OF THE FAILURE CASES OVER TREATMENT DISTRIBUTION SHIFTS

As shown in Figure 3 (a,c), with the shifts of the treatment interval, the estimation performance of DRNet and TARNet decline significantly. VCNet achieves $\infty$ estimation error when $h = 5$ partly because its hand-craft projection matrix can only process values near $[0, 1]$. Another problem brought by this assumption is the extrapolation dilemma, which can be seen in Figure 3(b). When training on $t \in [0, 1.75]$, these discrete approximation methods cannot transfer to new distribution $t \in (1.75, 2.0]$. These unseen treatments are rounded down to the nearest neighbors $t'$ in $T$ and be seemed the same as $t'$. We conduct ablation about the treatment embedding as in Table 6 in Appendix. Such a simple fix (VCNet+Embeddings) removes the demand on a fixed interval constraint to treatments and attains superior performance on both interpolation and extrapolation settings. The result clearly shows the pitfalls of hand-crafted feature mapping for TEE. We highlight that it is neglected by most existing works (Schwab et al., 2020; Nie et al., 2021; Shi et al., 2019; Guo et al., 2021). Extrapolation is still a challenging open problem. We can see that no existing work does well when training and test treatment intervals have big gaps. However, the empirical evidence validates the improved effectiveness of TransTEE that uses learnable embeddings to map continuous treatments to hidden representations.

Below we show the assumption that the value of treatments or dosages are in a fixed interval $[l, h]$ is sub-optimal and thus these methods get poor extrapolation results. For simplicity, we only consider a data sample has only one continuous treatment $t$ and the result is similar for continuous dosage.

**Proposition 2.** *Given a data sample* $(\mathbf{x}, t, y)$, *where* $\mathbf{x} \in \mathbb{R}^d, t \in [l, h], y \in \mathbb{R}$. *Assume* $\mu$ *is a L-Lipschitz function over* $(\mathbf{x}, t) \in \mathbb{R}^{d+1}$, *namely* $|\mu(\mathbf{u}) - \mu(\mathbf{v})| \leq L\|\mathbf{u} - \mathbf{v}\|$. *Partitioning* $[l, h]$ *uniformly into* $\delta$ *sub-interval, and then get* $T = \left[ l + \frac{h-l}{\delta} * 0, l + \frac{h-l}{\delta} * 1, ..., l + \frac{h-l}{\delta} * \delta \right]$. *Previous studies most rounding down a treatment* $t$ *to its nearest value in* $T$ *(either* $l + \left\lfloor \frac{t\delta}{h-l} \right\rfloor \frac{h-l}{\delta}$ *or* $l + \left\lceil \frac{t\delta}{h-l} \right\rceil \frac{h-l}{\delta}$ *) and use* $|T|$ *branches to approximate the entire continuum* $[l, h]$. *The approximation error can be bounded by*

$$
\begin{aligned}
\max & \left\{ \mu \left( \mathbf{x}, \left\lfloor \frac{t\delta}{h-l} \right\rfloor \frac{h-l}{\delta} \right) - \mu(\mathbf{x}, t), \mu \left( \mathbf{x}, \left\lceil \frac{t\delta}{h-l} \right\rceil \frac{h-l}{\delta} \right) - \mu(\mathbf{x}, t) \right\} \\
& \leq \max \left\{ L \left( \left| \left\lfloor \frac{t\delta}{h-l} \right\rfloor \frac{h-l}{\delta} - t \right| \right), L \left( \left| \left\lceil \frac{t\delta}{h-l} \right\rceil \frac{h-l}{\delta} - t \right| \right) \right\} \\
& \leq L \frac{h-l}{\delta}
\end{aligned}
\tag{11}
$$

The bound is affected by both the number of branches $\delta$ and treatment interval $[l, h]$. However, as far as we know, most previous works ignore the impacts of the treatment interval $[l, h]$ and adopt a simple but much stronger assumption that treatments are all in the interval $[0, 1]$ Nie et al. (2021) or a fixed interval Schwab et al. (2020). These observations well manifest the motivation of our general framework for TEE without the need for treatment-specific architectural designs.

Table 6: **Experimental results comparing NN-based methods on simulated datasets.** Numbers reported are AMSE of test data based on 100 repeats, and numbers after $\pm$ are the estimated standard deviation of the average value. For Extrapolation ($h = 2$), models are trained with $t \in [0, 1.75]$ and tested in $t \in [0, 2]$. For Extrapolation ($h = 5$), models are trained with $t \in [0, 4]$ and tested in $t \in [0, 5]$

| METHODS | VANILLA | VANILLA ($h = 5$) | EXTRAPOLATION ($h = 2$) | EXTRAPOLATION ($h = 5$) |
|---|---|---|---|---|
| TARNET (SHALIT ET AL., 2017) | $0.045 \pm 0.0009$ | $0.3864 \pm 0.04335$ | $0.0984 \pm 0.02315$ | $0.3647 \pm 0.03626$ |
| DRNET (SCHWAB ET AL., 2020) | $0.042 \pm 0.0009$ | $0.3871 \pm 0.03851$ | $0.0885 \pm 0.00094$ | $0.3647 \pm 0.03625$ |
| VCNET(NIE ET AL., 2021) | $0.018 \pm 0.0010$ | NAN | $0.0669 \pm 0.05227$ | NAN |
| VCNET+EMBEDDINGS | $0.013 \pm 0.00465$ | $0.0167 \pm 0.01150$ | $0.0118 \pm 0.00482$ | $0.0178 \pm 0.00887$ |

# E  ADDITIONAL EXPERIMENTAL SETUPS

All the assets (i.e., datasets and the codes for baselines) we use include a MIT license containing a copyright notice and this permission notice shall be included in all copies or substantial portions of the software. We conduct all the experiments on a machine with i7-8700K CPU, 32G RAM, and four Nvidia GeForce RTX2080Ti (10GB) GPU cards.

## E.1  DETAIL EVALUATION METRICS.

$$\text{AMSE}_{\mathcal{T}} = \frac{1}{N} \sum_{i=1}^{N} \int_{\mathcal{T}} \left[ \hat{f}(\mathbf{x}_i, t) - f(\mathbf{x}_i, t) \right] \pi(t) dt \tag{12}$$

$$\text{UPEHE@K} = \frac{1}{N} \sum_{i=1}^{N} \left[ \frac{1}{C_K^2} \sum_{t,t'} \left[ \hat{f}(\mathbf{x}_i, t, t') - f(\mathbf{x}_n, t, t') \right]^2 \right]$$

$$\text{WPEHE@K} = \frac{1}{N} \sum_{i=1}^{N} \left[ \frac{1}{C_K^2} \sum_{t,t'} \left[ \hat{f}(\mathbf{x}_i, t, t') - f(\mathbf{x}_i, t, t') \right]^2 p(t|\mathbf{x}) p(t'|\mathbf{x}) \right], \tag{13}$$

$$\text{AMSE}_{\mathcal{D}} = \frac{1}{NT} \sum_{i=1}^{N} \sum_{t=1}^{T} \int_{\mathcal{D}} \left[ \hat{f}(\mathbf{x}_i, t, s) - f(\mathbf{x}_n, t, s) \right] \pi(s) dt \tag{14}$$

## E.2  NETWORK STRUCTURE AND PARAMETER SETTING

Table. 7 and Table. 8 show the detail of TransTEE architecture and hyper-parameters. For all the synthetic and semi-synthetic datasets, we tune parameters based on 20 additional runs. In each run, we simulate data, randomly split it into training and testing, and use AMSE on testing data for evaluation. For fair comparisons, in all experiments, the model size of TransTEE is less than or similar to baselines.

Table 7: **Architecture details of TransTEE**, where $p$ is the number of covariates.

| Module | Covariates | Treatment |
|---|---|---|
| Embedding Layer | [Linear] | [Linear] |
| Output Size | Bsz $\times p \times$ #Emb | $bsz \times 1 \times$ # Emb |
| Self-Attention | $\begin{bmatrix} \text{Multi-head Att} \\ \text{BatchNorm} \\ \text{Linear} \\ \text{BatchNorm} \end{bmatrix} \times$ #Layers | $\begin{bmatrix} \text{Multi-head Att} \\ \text{BatchNorm} \\ \text{Linear} \\ \text{BatchNorm} \end{bmatrix} \times$ #Layers |
| Output Size | Bsz $\times p \times$ #Emb | Bsz $\times 1 \times$ #Emb |
| Cross-Attention | $\begin{bmatrix} \text{Multi-head Att} \\ \text{BatchNorm} \\ \text{Linear} \\ \text{BatchNorm} \end{bmatrix} \times$ #Layers | |
| Output Size | Bsz $\times 1 \times$ #Emb | |
| Projection Layer | [Linear] | |
| Output Size | Bsz $\times 1$ | |

## E.3  SIMULATION DETAILS.

**Synthetic Dataset** (Nie et al., 2021). The synthetic dataset contains 500 training points and 200 testing points. Data is generated as follows: $x_j \sim \text{Unif}[0, 1]$, where $x_j$ is the $j$-th dimension of

Table 8: **Hyper-parameters on different datasets**. Bsz indicates the batch size, # Emb indicates the embedding dimension, Lr. S indicates the scheduler of the learning rate (Cos is the cosine annealing Learning rate).

| Dataset | Bsz | # Emb | # Layers | # Heads | Lr | Lr. S |
|---------|-----|-------|----------|---------|--------|-------|
| Simu | 500 | 10 | 1 | 2 | 0.01 | Cos |
| IHDP | 128 | 10 | 1 | 2 | 0.0005 | Cos |
| News | 256 | 10 | 1 | 2 | 0.01 | Cos |
| SW | 500 | 16 | 1 | 2 | 0.01 | None |
| TCGA | 1000 | 48 | 3 | 4 | 0.01 | None |

$x \in \mathbb{R}^6$, and

$$\tilde{t}|x = \frac{10\sin\left(\max(x_1, x_2, x_3)\right) + \max(x_3, x_4, x_5)^3}{1 + (x_1 + x_5)^2} + \sin(0.5x_3)\left(1 + \exp(x_4 - 0.5x_3)\right) +$$

$$x_3^2 + 2\sin(x_4) + 2x_5 - 6.5 + \mathcal{N}(0, 0.25)$$

$$y|x, t = \cos(2\pi(t - 0.5))\left(t^2 + \frac{4\max(x_1, x_6)^3}{1 + 2x_3^2}\right) + \mathcal{N}(0, 0.25)$$

where $t = (1 + \exp(-\tilde{t}))^{-1}$.

for treatment in $[0, h]$, we revised it to $t = (1 + \exp - \tilde{t})^{-1} * h$,

**IHDP** (Hill, 2011) is a semi-synthetic dataset containing 25 covariates, 747 observations and binary treatments. For treatments in $[0, 1]$, we follow VCNet (Nie et al., 2021) and generate treatments and responses by:

$$\tilde{t}|x = \frac{2x_1}{1 + x_2} + \frac{2\max(x_3, x_5, x_6)}{0.2 + \min(x_3, x_5, x_6)} + 2\tanh\left(5\frac{\sum_{i \in S_{dis,2}}(x_i - c_2)}{|S_{dis,2}|} - 4 + \mathcal{N}(0, 0.25)\right)$$

$$y|x, t = \frac{\sin(3\pi t)}{1.2 - t}\left(\tanh\left(5\frac{\sum_{i \in S_{dis,1}}(x_i - c_1)}{|S_{dis,1}|}\right) + \frac{\exp(0.2(x_1 - x_6))}{0.5 + 5\min(x_2, x_3, x_5)}\right) + \mathcal{N}(0, 0.25),$$

where $t = (1 + \exp(-\tilde{t}))^{-1}$, $S_{con} = \{1, 2, 3, 5, 6\}$ is the index set of continuous features, $S_{dis,1} = \{4, 7, 8, 9, 10, 11, 12, 13, 14, 15\}$, $S_{dis,2} = \{16, 17, 18, 19, 20, 21, 22, 23, 24, 25\}$ and $S_{dis,1} \bigcup S_{dis,2} = [25] - S_{con}$. Here $c_1 = \mathbb{E}\left[\frac{\sum_{i \in S_{dis,1}} x_i}{|S_{dis,1}|}\right], c_2 = \mathbb{E}\left[\frac{\sum_{i \in S_{dis,2}} x_i}{|S_{dis,2}|}\right]$. To allow comparison on various treatment intervals $t \in [0, h]$, treatments and responses are generated by:

$$t = (1 + \exp(-\tilde{t}))^{-1} * h$$

$$y|x, t = \frac{\sin(3\pi t/h)}{1.2 - t/h}\left(\tanh\left(5\frac{\sum_{i \in S_{dis,1}}(x_i - c_1)}{|S_{dis,1}|}\right) + \frac{\exp(0.2(x_1 - x_6))}{0.5 + 5\min(x_2, x_3, x_5)}\right) + \mathcal{N}(0, 0.25),$$

where the orange part is the only different compared to the generalization of vanilla IHDP dataset ($h = 1$). Note that $S_{dis,1}$ only impacts outcome that serves to be noisy covariates; $S_{dis,2}$ contains pre-treatment covariates that only impact treatments, which also serves to be instrumental variables. This allows us to observe the improvement using TransTEE when noisy covariates exist. Following (Hill, 2011) covariates are standardized with mean 0 and standard deviation 1.

**News.** The News dataset consists of 3000 randomly sampled news items from the NY Times corpus (Newman, 2008) and was originally introduced as a benchmark in the binary treatment setting. We generate the treatment and outcome in a similar way as (Nie et al., 2021) but with a dynamic range or treatment intervals $[0, h]$. We first generate $v_1', v_2', v_3' \sim \mathcal{N}(0, 1)$ and then set $v_i = v_i'/\|v_i'\|_2; i \in \{1, 2, 3\}$. Given $x$, we generate $t$ from Beta $\left(2, \left|\frac{v_3^\top x}{2v_2^\top x}\right|\right) * h$. And we generate the outcome by

$$y'|x, t = \exp\left(\frac{v_2^\top x}{v_3^\top x} - 0.3\right)$$

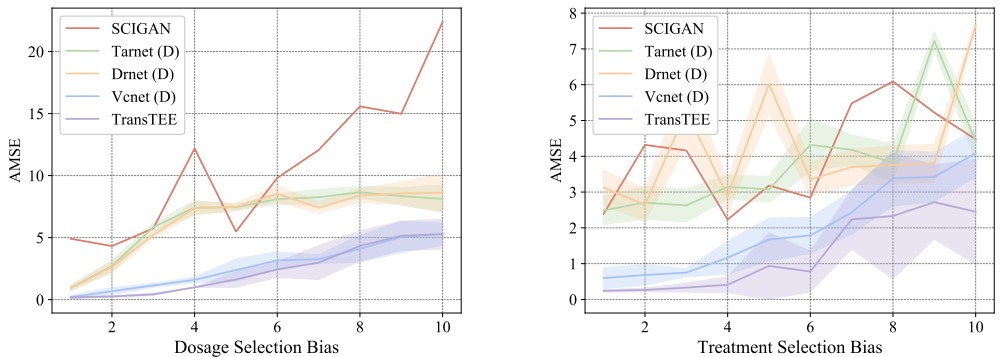

(a) Performance with different dosage selection bias.    (b) Performance with different treatment selection bias.

Figure 7: Performance of five methods on TCGA (D) dataset with varying bias levels.

$$y|x, t = 2(\max(-2, \min(2, y')) + 20v_1^\top x) * \left(4(t - 0.5)^2 + \sin\left(\frac{\pi}{2}t\right)\right) + \mathcal{N}(0, 0.5)$$

**TCGA (D)** (Bica et al., 2020b) We obtain covariates $x$ from a real dataset *The Cancer Genomic Atlas (TCGA)* and consider 3 treatments, where each treatment is accompanied by one dosage and a set of parameters, $v_1^t, v_2^t, v_3^t$. For each run, we randomly sample a vector, $u_i^t \sim \mathcal{N}(0, 1)$ and then set $v_i^t = u_i^t / \|u_i^t\|$ where $\|\cdot\|$ is Euclidean norm. The shape of the response curve for each treatment, $f_t(x, s)$ is given in Table 9. We add $\epsilon \sim \mathcal{N}(0, 0.2)$ noise to the outcomes. Interventions are assigned by sampling a dosage, $d_t$, for each treatment from a beta distribution, $d_t|x \sim \text{Beta}(\alpha, \beta_t)$. $\alpha \geq 1$ controls the dosage selection bias ($\alpha = 1$ gives the uniform distribution). $\beta_t = \frac{\alpha - 1}{s_t^*} + 2 - \alpha$, where $s_t^*$ is the optimal dosage[2] for treatment $t$. We then assign a treatment according to $t_f|x \sim \text{Categorical}(\text{Softmax}(\kappa f(x, s_t)))$ where increasing $\kappa$ increases selection bias, and $\kappa = 0$ leads to random assignments. The factual intervention is given by $(t_f, s_{t_f})$. Unless otherwise specified, we set $\kappa = 2$ and $\alpha = 2$.

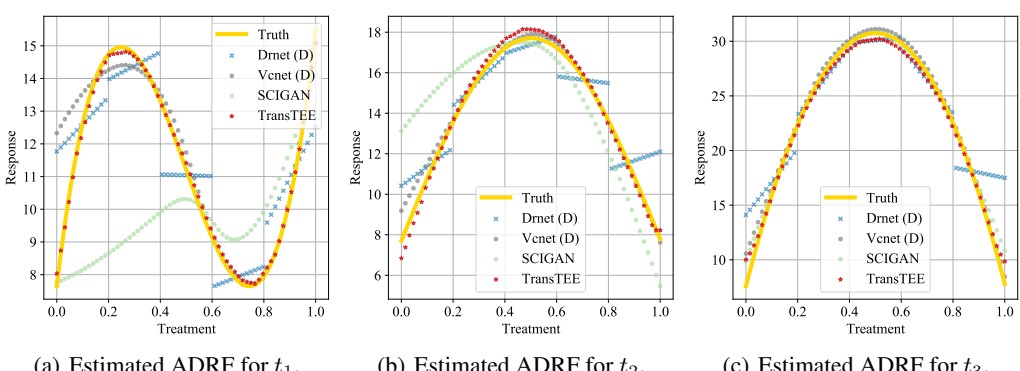

(a) Estimated ADRF for $t_1$.    (b) Estimated ADRF for $t_2$.    (c) Estimated ADRF for $t_3$.

Figure 8: **Estimated ADRF** on the test set from a typical run of DRNet (D), TARNet (D), VCNet (D), and SCIGAN. All of these methods are well optimized. TransTEE can well estimate the dosage-response curve for all treatments.

For structural treatments, we first define the **Baseline effect** (Bica et al., 2020b). For each run of the experiment, we randomly sample a vector $u_0 \sim \text{Unif}[0, 1]$, and set $v_0 = u_0 / \|u_o\|$, where $\|\cdot\|$ is the Euclidean norm. The baseline effect is defined as

$$\mu_0(x) = v_0^\top x$$

---

[2]For symmetry, if $s_t^* = 0$, we sample $s_t^*$ from $1 - \text{Beta}(\alpha, \beta_t)$ where $\beta_t$ is set as though $s_t^* = 1$.

Table 9: **Dose response curves used to generate semi-synthetic outcomes for patient features** $x$. In the experiments, we set $C = 10$. $v_1^t, v_2^t, v_3^t$ are the parameters associated with each treatment $t$.

| Treatment | Dose-Response | Optimal dosage |
|---|---|---|
| 1 | $f_1(x, s) = C\left((v_1^1)^\top x + 12(v_3^1)^\top x s - 12(v_3^1)^\top x s^2\right)$ | $s_1^* = \frac{(v_2^1)^\top x}{2(v_3^1)^\top x}$ |
| 2 | $f_2(x, s) = C\left((v_1^2)^\top x + \sin\left(\pi(\frac{v_2^{2\top} x}{v_3^{2\top} x} s)\right)\right)$ | $s_2^* = \frac{(v_3^3)^\top x}{2(v_2^2)^\top x}$ |
| 3 | $f_3(x, s) = C\left((v_1^3)^\top x + 12 s(s-b)^2, \text{where } b = 0.75\frac{(v_2^3)^\top x}{(v_3^3)^\top x}\right)$ | $\frac{b}{3}$ if $b \geq 0.75$ else 1 |

**Small-World** (Kaddour et al., 2021). 20-dimensional multivariate covariates are uniformly sampled according to $x_i \sim \text{Unif}[-1, 1]$. There are $1,000$ units in in-sample dataset, and $500$ in the out-sample one. *Graph interventions* For each graph intervention, a number of nodes between 10 and 120 are uniformly sampled, the number of neighbors for each node is between 3 and 8, and the probability of rewiring each edge is between $0.1$ and $1$. Watts–Strogatz small-world graphs are repeatedly generated until a connected one is get. Each vertex has one feature, i.e. its degree centrality. A graph's node connectivity is denoted as $\nu(\mathcal{G})$ and its average shortest path length as $\ell(\mathcal{G})$. Similar for the baseline effect, two randomly sampled vectors $v_\nu, v_\ell$ are generated. Then, given an assigned graph treatment $\mathcal{G}$ and a covariate vector $x$, the *outcome* is generated by

$$y = 100\mu_0(x) + 0.2\nu(\mathcal{G})^2 \cdot v_\nu^\top x + \ell(\mathcal{G}) \cdot v_\ell^\top x + \epsilon, \epsilon \sim \mathcal{N}(0, 1)$$

**TCGA (S)** (Kaddour et al., 2021) We use $9,659$ gene expression measurements of cancer patients for covariates. The in-sample and datasets consist of $5,000$ units and the out-sample one of $4,659$ units, respectively. Each unit is a covariate vector $x \in \mathbb{R}^{4000}$ and these units are split randomly into in- and out-sample datasets in each run randomly. For each covariate vector $x$, its 8-dimensional PCA components $x^{\text{PCA}} \in \mathbb{R}^8$ is computed. *Graph interventions* We randomly sample $10,000$ molecules from the Quantum Machine 9 (QM9) dataset (Ramakrishnan et al., 2014) (with 133k molecules in total) in each run. We create a relational graph, where each node corresponds to an atom and consists of 78 atom features. We label each edge corresponding to the chemical bond types, e.g., single, double, triple, and aromatic bonds. We collect 8 molecule properties $mu, alpha, homo, lumo, gap, r2, zpve, u0$ in a vector $z \in \mathbb{R}^8$, which is denoted as the the assigned molecule treatment. Finally, we generate *outcomes* by

$$y = 10\mu_0(x) + 0.01 z^\top x^{\text{PCA}} + \epsilon, \epsilon \sim \mathcal{N}(0, 1)$$

**Enriched Equity Evaluation Corpus (EEEC)** (Feder et al., 2021) consists of $33,738$ English sentences and the label of each sentence is the mood state it conveys. The task is also known as Profile of Mood States (POMS). Each sentence in the dataset is created using one of 42 templates, with placeholders for a person's name and the emotion, e.g., *"<Person> made me feel <emotional state word>."*. A list of common names that are tagged as male or female, and as African-American or European will be used to fill the placeholder (*<Person>*). One of four possible mood states: *Anger*, *Sadness*, *Fear* and *Joy* is used to fill the emotion placeholder. Hence, EEEC has two kinds of counterfactual examples, which are *Gender* and *Race*. For the *Gender* case, it changes the name and the *Gender* pronouns in the example and switches them, such that for the original example: *"It was totally unexpected, but Roger made me feel pessimistic."* it will have the counterfactual example: *"It was totally unexpected, but Amanda made me feel pessimistic."* For the *Race* concept, it creates counterfactuals such that for the original example *"Josh made me feel uneasiness for the first time ever in my life."*, the counterfactual example is: *"Darnell made me feel uneasiness for the first time ever in my life."* For each counterfactual example, the person's name is taken at random from the pre-existing list corresponding to its type.

## F  ADDITIONAL EXPERIMENTAL RESULTS

### F.1  COMPARISION BETWEEN TRANSTEE AND ANU (XU ET AL., 2022)

We implement ANU and evaluate it in the same settings and show that is inferior compared to the proposed TransTEE as follows. Specifically, we compare the attentive neural uplift model (ANU) (Xu

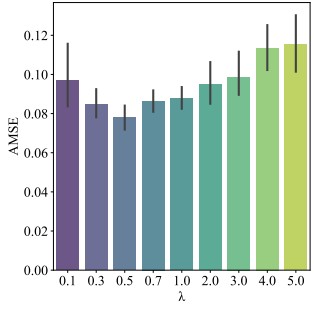

(a) Ablation study of PTR.

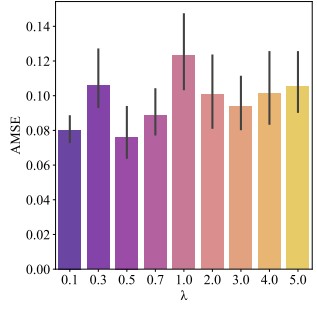

(b) Ablation study of TR.

Figure 9: Ablation study of the balanced weight for treatment regularization on the IHDP dataset.

Table 10: Comparision between TransTEE and ANU (Xu et al., 2022) on the IHDP dataset.

| Methods | Vanilla (Binary) | Vanilla (h = 1) | Extrapolation (h = 2) |
|---|---|---|---|
| DRNet | $0.3543 \pm 0.6062$ | $2.1549 \pm 1.04483$ | $11.071 \pm 0.9938$ |
| VCNet | $0.2098 \pm 0.18236$ | $0.7800 \pm 0.6148$ | NAN |
| ANU (Xu et al., 2022) | $0.1482 \pm 0.17362$ | $0.2147 \pm 0.32451$ | $0.4244 \pm 0.19832$ |
| TransTEE | $0.0983 \pm 0.15384$ | $0.1151 \pm 0.1028$ | $0.2745 \pm 0.1497$ |

et al., 2022) with ours in the following two settings. (1) IHDP dataset in Table 10 in the main manuscript. We adjust the layers of ANU such that the total parameters of ANU and TransTEE are similar. The result is shown in the following table. With the usage of treatment embeddings, ANU is shown to be more robust than VCNet and DRNet when a treatment shift occurs. However, in both the binary treatment setting and continuous treatment settings, TransTEE performs better than ANU.

(2) We further evaluate the real-world utility of ANU (Xu et al., 2022) and the experimental setting is detailed in Section 5.4 in the main paper. Covariates here are long sentences. Thanks to the use of self-attention modules, TransTEE can achieve better estimation results compared to baselines (Table 11). For AHU, no self-attention layer is applied, and the final estimation is inaccurate, which verifies the superiority of the proposed framework.

## F.2 ADDITIONAL NUMERICAL RESULTS AND ABLATION STUDIES

**Choice of the balancing weight for treatment regularization.** To understand the effect of propensity score modeling, we conduct an ablation study on the balancing weights of both TR and PTR. Figure 9 presents the results of the experiments on the IHDP dataset. The main observation is that both TR and PTR with a proper regularization strength consistently improve estimation compared to TransTEE without regularization. The best performers are achieved when $\lambda$ is $0.5$ for both two methods, which shows that the best balancing parameter ($0.5$ on our experiments.) for these two regularization terms should be searched carefully. Besides, training both the treatment predictor and the feature encoder simultaneously in a zero-sum game is difficult and sometimes unstable (shown in Figure 9 right)

**Robustness to noisy covariates.** We manipulate $S_{dis,1}, S_{dis,2}$ to generate datasets with different noisy covariates, e.g., when the *number of covariates that only influence the outcome* is 6,

Table 11: Comparision between TransTEE and ANU (Xu et al., 2022) on the IHDP dataset.

| | Correlation/Representation Based Baselines | | | | | Treatment Effect Estimators | | | |
|---|---|---|---|---|---|---|---|---|---|
| TC | $ATE_{GT}$ | TReATE | CONEXP | INLP | CausalBERT | TarNet | DRNet | ANU | TransTEE |
| Gender | 0.086 | 0.125 | 0.02 | 0.313 | 0.179 | 0.0067 | 0.0088 | 0.184 | 0.013 |
| Race | 0.014 | 0.046 | 0.08 | 0.591 | 0.213 | 0.005 | 0.006 | 0.093 | 0.0174 |

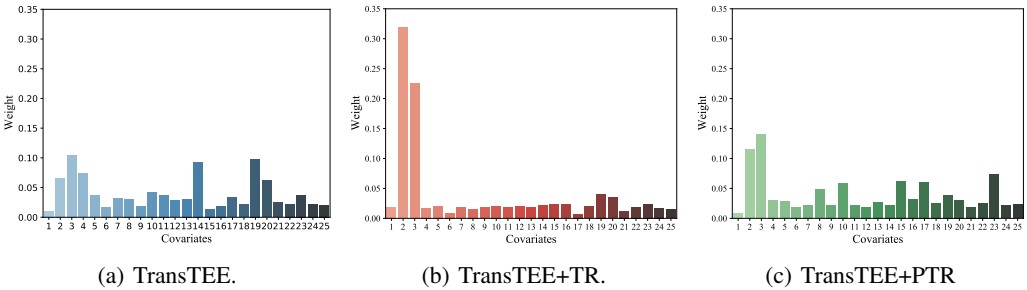

Figure 10: The distribution of learned weights for the cross-attention module on the IHDP dataset of different models.

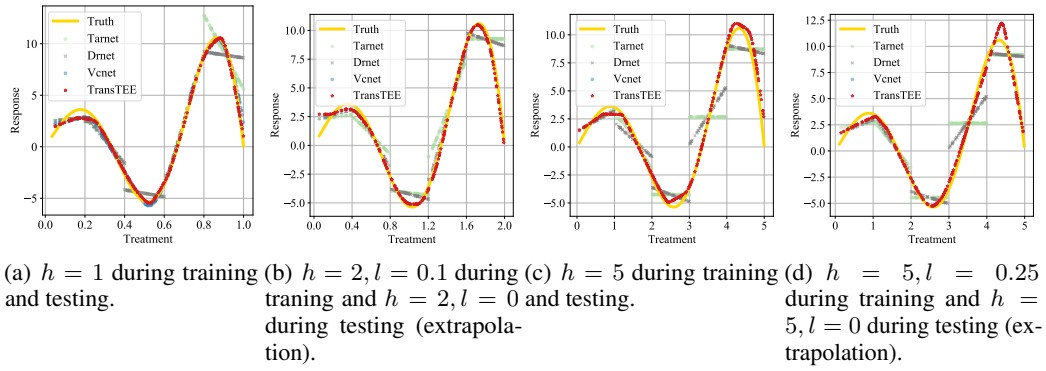

(a) $h = 1$ during training and testing.

(b) $h = 2, l = 0.1$ during traning and $h = 2, l = 0$ and testing. during testing (extrapolation).

(c) $h = 5$ during training and $h = 2, l = 0$ and testing.

(d) $h = 5, l = 0.25$ during training and $h = 5, l = 0$ during testing (extrapolation).

Figure 11: **Estimated ADRF** on test set from a typical run of TarNet (Shalit et al., 2017), DR-Net (Schwab et al., 2020), VCNet (Nie et al., 2021) and ours on IHDP dataset. All of these methods are well optimized. (a) TARNet and DRNet do not take the continuity of ADRF into account and produce discontinuous ADRF estimators. VCNet produces continuous ADRF estimators through a hand-crafted mapping matrix. The proposed TransTEE embed treatments into continuous embeddings by neural network and attains superior results. (b,d) When training with $0.1 \leq t \leq 2.0$ and $0.25 \leq t \leq 5.0$. TARNet and DRNet cannot extrapolate to distributions with $0 < t \leq 2.0$ and $0 \leq t \leq 5.0$. (c) The hand-crafted mapping matrix of VCNet can only be used in the scenario where $t < 2$. Otherwise, VCNet cannot converge and incur an infinite loss. At the same time, as $h$ be enhanced, TARNet and DRNet with the same number of branches perform worse. TransTEE needs not to know $h$ in advance and extrapolates well.

$S_{dis,1} = \{4\}$, and $S_{dis,2} = \{7, 8, 9, 10, 11, 12, 13, 14, 15, 16, 17, 18, 19, 20, 21, 22, 23, 24, 25\}$, when the *number of covariates that influence the outcome* is 24, $S_{dis,1} = \{4, 7, 8, 9, 10, 11, 12, 13, 14, 15, 16, 17, 18, 19, 20, 21, 22, 23, 24, \}$, and $S_{dis,2} = \{25\}$. Figure Figure 4(b) shows that, as the number of covariates that only influence the outcome increases, both TARNet and DRNet become better estimators, however, VCNet performs worse and even inferior to TARNet and DRNet when the number is large than 16. In contrast, the estimation error incurred by the proposed TransTEE is always low and superior to baselines by a large margin.

**Comparison of MLE or adversarial propensity score modeling on the propensity score.** Seeing results in Table 2, additionally combine TransTEE with maximum likelihood training of $\pi(t|\mathbf{x})$ does provide some performance gains. However, an adversarially trained $\pi$-model can be significantly better, especially for extrapolation settings. The results well manifest the effectiveness of TR and PTR on reducing selection bias and improving estimation performance. In fact, approaches like TMLE are not robust if the initial estimator is poor Shi et al. (2019).

**Training dynamics comparison of different regularization terms.** Here we compare four regularization terms, which are TransTEE with no regularization, TransTEE+TR, TransTEE+PTR, and TransTEE+MTL. TransTEE+MTL is a simple **M**ulti-**T**ask **L**earning strategy, which uses

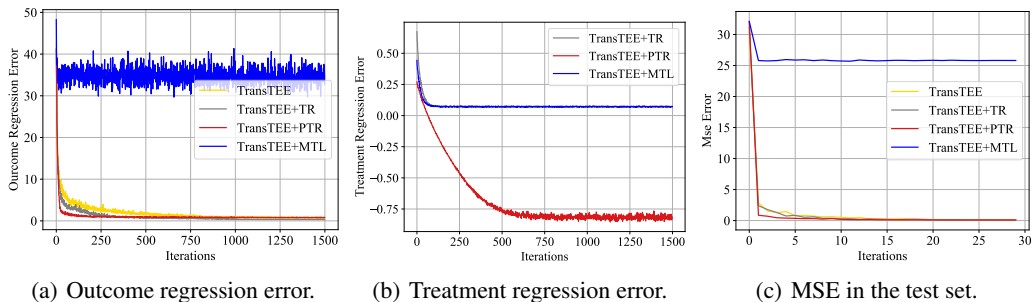

| (a) Outcome regression error. | (b) Treatment regression error. | (c) MSE in the test set. |

Figure 12: **Training dynamics of TransTEE** on IHDP dataset with various regularization terms, where the total training iteration is $1,500$ and (c) is evaluated on the test set per 50 training iterations.

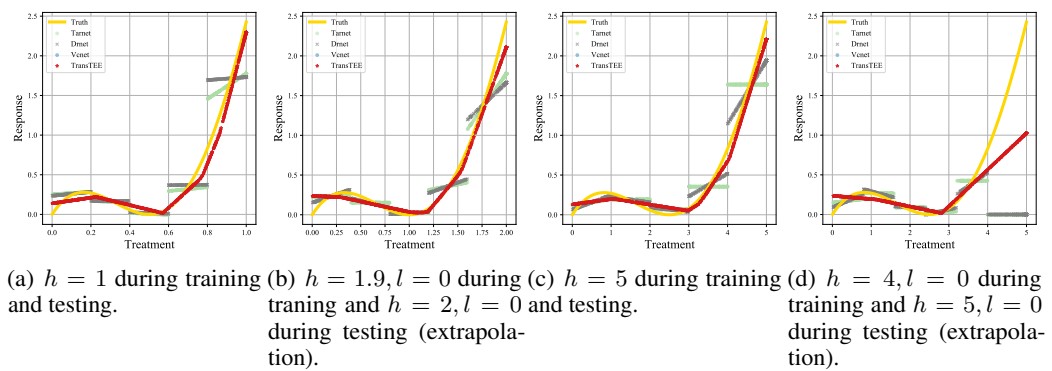

(a) $h = 1$ during training and testing.

(b) $h = 1.9, l = 0$ during traning and $h = 2, l = 0$ and testing.

(c) $h = 5$ during training during testing (extrapolation).

(d) $h = 4, l = 0$ during training and $h = 5, l = 0$ during testing (extrapolation).

Figure 13: **Estimated ADRF** on the test set from a typical run of TarNet (Shalit et al., 2017), DRNet (Schwab et al., 2020), VCNet (Nie et al., 2021) and ours on News dataset. All of these methods are well optimized. Suppose $t \in [l, h]$. (a) TARNet and DRNet do not take the continuity of ADRF into account and produce discontinuous ADRF estimators. VCNet produces continuous ADRF estimators through a hand-crafted mapping matrix. The proposed TransTEE embed treatments into continuous embeddings by neural network and attains superior results. (b,d) When training with $0 \le t \le 1.9$ and $0 \le t \le 4.0$. TARNet and DRNet cannot extrapolate to distributions with $0 < t \le 2.0$ and $0 \le t \le 5.0$. (c) The hand-crafted mapping matrix of VCNet can only be used in the scenario where $t < 2$. Otherwise, VCNet cannot converge and incur an infinite loss. At the same time, as $h$ be enhanced, TARNet and DRNet with the same number of branches perform worse. TransTEE needs not know $h$ in advance and extrapolates well.

$\mathcal{L}_\theta(\mathbf{x}, y, t) + \mathcal{L}_\phi^{TR}(\mathbf{x}, t)$ during training without an adversarial game. As shown in Figure 12, without adversarial training, TransTEE+MTL quickly attains low treatment estimation error but further oscillate and converge with a high error, and both the outcome regression error and MSE in the test set remain high. In contrast, TR and PTR make TransTEE converge faster and attain lower test MSE. Overall, PTR consistently works the best and its low treatment regression error shows that $\pi_\phi(t|\mathbf{x})$ estimates an accurate propensity score.

### F.3 SHOWCASE OF SENTENCES AND COUNTERFACTUAL COUNTERPARTS WITH THE MAXIMAL/MINIMAL ATES.

Table 14 showcases the top-10 samples with the maximal/ minimal ATEs. Interestingly, we can see most sentences with a large ATE have similar patterns, that is "$< clause >$, *but/and* $< Person >$ *made me feel* $< Adj >$". Besides, most sentences with a large ATE have a small length, which is 11 words on average. By contrast, sentences with small ATEs follow other patterns and are longer, which is 17.6 on average. Consider the effect of *Race*, Table 15 showcases the top-10 samples. Similarly, there are also some dominant patterns that have pretty high or low ATEs and the average length of sentences with high ATEs is smaller than sentences with low ATEs ($12 \ vs \ 14.7$). Besides, the position

Table 12: **Experimental results comparing neural network based methods on the News datasets.** Numbers reported are based on 20 repeats, and numbers after $\pm$ are the estimated standard deviation of the average value. For Extrapolation ($h = 2$), models are trained with $t \in [0, 1.9]$ and tested in $t \in [0, 2]$. For For Extrapolation ($h = 5$), models are trained with $t \in [0, 4.5]$ and tested in $t \in [0, 5]$

| METHODS | VANILLA | VANILLA ($h = 5$) | EXTRAPOLATION ($h = 2$) | EXTRAPOLATION ($h = 5$) |
|---|---|---|---|---|
| TARNET | $0.082 \pm 0.019$ | $0.956 \pm 0.041$ | $0.716 \pm 0.038$ | $0.847 \pm 0.053$ |
| DRNET | $0.083 \pm 0.032$ | $0.956 \pm 0.041$ | $0.703 \pm 0.038$ | $0.834 \pm 0.053$ |
| VCNET | $0.013 \pm 0.005$ | NAN | NAN | NAN |
| TRANSTEE | $\mathbf{0.010 \pm 0.004}$ | $0.017 \pm 0.008$ | $0.024 \pm 0.017$ | $0.029 \pm 0.019$ |
| TRANSTEE+TR | $0.011 \pm 0.003$ | $0.016 \pm 0.008$ | $\mathbf{0.019 \pm 0.008}$ | $\mathbf{0.028 \pm 0.002}$ |
| TRANSTEE+PTR | $0.011 \pm 0.004$ | $\mathbf{0.014 \pm 0.007}$ | $0.022 \pm 0.008$ | $0.029 \pm 0.016$ |

of perturbation words (the name from a specific race) for sentences with the maximal/minimal ATEs is totally different, which is at the beginning for the former and at the middle for the latter. Namely, TransTEE helps us mitigate spurious correlations that exist in model prediction, e.g., length of sentences, the position of perturbation words, certain sentence patterns and is useful in mitigating undesirable bias ingrained in the data. Besides, a well-optimized TransTEE is able to estimate the effect of every sentence and is of great benefit for model interpretation and analysis especially under high inference latency.

## G  REMARKS ON INTERPRETABILITY

It is fundamentally hard to evaluate the interpretability even for supervised learners, as the evaluation crucially depends on specific models, tasks, and input spaces (Jacovi & Goldberg, 2020). TransTEE provide an initial step to promote causal inference model interpretability. We can see from the experimental results in fig. 4(a), 4(b), and fig. 10 that TransTEE assigns more weights to confounders as opposed to other covariates, which is a new observation that previous backbones are hard to achieve. We see that explaining causal inference models in this way - using the feature importance scores for each covariate can be used for benchmarking treatment effect estimators (Crabbé et al., 2022).

Table 13: **Error of CATE estimation for all methods, measured by WPEHE@2-10.** Results are averaged over 5 trials, ± denotes std error. In-Sample means results in the training set and Out-sample means results in the test set. (The baseline results are reproduced using the official code of (Kaddour et al., 2021) in a consistent experimental environment, which can be slightly different than the results reported in (Kaddour et al., 2021))

| Method | SW | | TCGA (Bias=0.1) | | TCGA (Bias=0.3) | | TCGA (Bias=0.5) | |
|---|---|---|---|---|---|---|---|---|
| | In-sample | Out-sample | In-sample | Out-sample | In-sample | Out-sample | In-sample | Out-sample |
| | | | | WPEHE@2 | | | | |
| Zero | 41.72 ± 0.00 | 49.69 ± 0.00 | 13.93 ± 0.00 | 13.13 ± 0.00 | 13.93 ± 0.00 | 13.13 ± 0.00 | 13.93 ± 0.00 | 13.61 ± 0.00 |
| GNN | 17.38 ± 0.01 | 24.53 ± 0.01 | 10.90 ± 7.71 | 10.91 ± 7.71 | 13.58 ± 0.18 | 13.22 ± 0.18 | 12.86 ± 0.38 | 14.62 ± 0.91 |
| GraphITE | 17.37 ± 0.01 | 24.56 ± 0.02 | 15.04 ± 0.20 | 14.96 ± 0.30 | 13.49 ± 0.23 | 13.70 ± 0.52 | 12.41 ± 0.02 | 14.38 ± 0.30 |
| SIN | 15.79 ± 1.72 | 28.78 ± 4.54 | 46.47 ± 2.19 | 54.41 ± 7.81 | 7.93 ± 0.79 | 11.04 ± 1.52 | 10.31 ± 0.93 | 14.09 ± 2.14 |
| TransTEE | **14.74 ± 0.09** | **21.78 ± 1.07** | **9.07 ± 2.15** | **9.33 ± 2.13** | **7.54 ± 3.60** | **8.37 ± 3.64** | **9.52 ± 3.59** | **10.10 ± 3.79** |
| | | | | WPEHE@3 | | | | |
| Zero | 40.75 ± 0.00 | 43.76 ± 0.00 | 13.93 ± 0.00 | 13.61 ± 0.00 | 13.93 ± 0.00 | 13.61 ± 0.00 | 13.61 ± 0.00 | 14.14 ± 0.00 |
| GNN | 18.26 ± 0.00 | 20.91 ± 0.01 | 10.75 ± 7.60 | 10.91 ± 7.72 | 13.63 ± 0.18 | 13.58 ± 0.19 | 12.92 ± 0.33 | 15.29 ± 1.04 |
| GraphITE | 18.27 ± 0.01 | 20.95 ± 0.02 | 14.88 ± 0.19 | 15.12 ± 0.29 | 13.49 ± 0.22 | 14.19 ± 0.43 | 12.56 ± 0.01 | 15.18 ± 0.31 |
| SIN | 18.15 ± 1.97 | 23.62 ± 3.93 | 45.29 ± 2.33 | 53.72 ± 8.09 | 7.94 ± 0.75 | 11.53 ± 1.59 | 10.89 ± 1.07 | 14.27 ± 1.92 |
| TransTEE | **15.30 ± 1.12** | **18.73 ± 2.09** | **9.07 ± 2.02** | **9.58 ± 2.04** | **7.58 ± 3.62** | **8.65 ± 3.75** | **9.64 ± 3.56** | **10.59 ± 3.88** |
| | | | | WPEHE@4 | | | | |
| Zero | 45.74 ± 0.00 | 44.95 ± 0.00 | 14.14 ± 0.00 | 13.75 ± 0.00 | 14.14 ± 0.00 | 13.75 ± 0.00 | 13.75 ± 0.00 | 14.31 ± 0.00 |
| GNN | 22.09 ± 0.01 | 23.01 ± 0.01 | 10.87 ± 7.69 | 10.88 ± 7.69 | 13.87 ± 0.18 | 13.71 ± 0.19 | 13.13 ± 0.34 | 15.47 ± 1.05 |
| GraphITE | 22.12 ± 0.00 | 23.03 ± 0.02 | 15.05 ± 0.18 | 15.14 ± 0.28 | 13.64 ± 0.20 | 14.30 ± 0.35 | 12.77 ± 0.02 | 15.38 ± 0.30 |
| SIN | 22.14 ± 2.30 | 23.70 ± 3.67 | 44.72 ± 2.35 | 53.12 ± 8.09 | 7.99 ± 0.73 | 11.66 ± 1.59 | 11.38 ± 1.04 | 14.37 ± 1.83 |
| TransTEE | **18.99 ± 0.83** | **19.65 ± 1.97** | **9.09 ± 1.97** | **9.66 ± 2.01** | **7.67 ± 3.70** | **8.71 ± 3.78** | **9.78 ± 3.63** | **10.74 ± 3.91** |
| | | | | WPEHE@5 | | | | |
| Zero | 49.19 ± 0.00 | 45.96 ± 0.00 | 14.31 ± 0.00 | 13.95 ± 0.00 | 14.31 ± 0.00 | 13.95 ± 0.00 | 13.95 ± 0.00 | 14.47 ± 0.00 |
| GNN | 24.18 ± 0.01 | 24.20 ± 0.01 | 10.99 ± 7.77 | 10.97 ± 7.76 | 13.98 ± 0.17 | 13.92 ± 0.18 | 13.31 ± 0.37 | 15.67 ± 1.05 |
| GraphITE | 24.22 ± 0.01 | 24.22 ± 0.03 | 15.24 ± 0.19 | 15.29 ± 0.28 | 13.68 ± 0.17 | 14.37 ± 0.37 | 12.95 ± 0.03 | 15.59 ± 0.30 |
| SIN | 25.48 ± 3.02 | 25.44 ± 3.50 | 44.55 ± 2.35 | 52.78 ± 8.04 | 8.10 ± 0.75 | 11.76 ± 1.59 | 11.75 ± 1.22 | 14.59 ± 1.84 |
| TransTEE | **20.16 ± 0.42** | **21.08 ± 1.78** | **9.17 ± 1.96** | **9.72 ± 2.00** | **7.76 ± 3.75** | **8.80 ± 3.82** | **9.91 ± 3.66** | **10.89 ± 3.94** |
| | | | | WPEHE@6 | | | | |
| Zero | 49.95 ± 0.00 | 50.10 ± 0.00 | 14.47 ± 0.00 | 14.04 ± 0.00 | 14.47 ± 0.00 | 14.04 ± 0.00 | 14.04 ± 0.00 | 14.53 ± 0.00 |
| GNN | 25.13 ± 0.00 | 26.93 ± 0.01 | 11.11 ± 7.86 | 11.02 ± 7.79 | 14.07 ± 0.22 | 14.11 ± 0.18 | 13.45 ± 0.38 | 15.76 ± 1.04 |
| GraphITE | 25.17 ± 0.02 | 26.94 ± 0.02 | 15.40 ± 0.19 | 15.37 ± 0.28 | 13.74 ± 0.12 | 14.58 ± 0.38 | 13.09 ± 0.04 | 15.68 ± 0.29 |
| SIN | 27.07 ± 2.98 | 28.11 ± 3.51 | 44.48 ± 2.35 | 52.54 ± 7.99 | 8.22 ± 0.75 | 11.82 ± 1.58 | 11.97 ± 1.19 | 14.74 ± 1.86 |
| TransTEE | **21.32 ± 0.79** | **22.99 ± 1.43** | **9.23 ± 1.95** | **9.77 ± 1.99** | **7.80 ± 3.83** | **8.84 ± 3.89** | **10.01 ± 3.70** | **10.96 ± 3.95** |
| | | | | WPEHE@7 | | | | |
| Zero | 55.40 ± 0.00 | 58.42 ± 0.00 | 14.53 ± 0.00 | 14.09 ± 0.00 | 14.53 ± 0.00 | 14.09 ± 0.00 | 14.53 ± 0.00 | 14.09 ± 0.00 |
| GNN | 29.30 ± 0.03 | 32.15 ± 0.03 | 11.16 ± 7.89 | 11.06 ± 7.82 | 14.12 ± 0.21 | 14.14 ± 0.18 | 13.51 ± 0.38 | 15.81 ± 1.03 |
| GraphITE | 29.34 ± 0.01 | 32.16 ± 0.01 | 15.47 ± 0.19 | 15.42 ± 0.28 | 13.97 ± 0.08 | 14.69 ± 0.40 | 13.16 ± 0.04 | 15.74 ± 0.29 |
| SIN | 31.07 ± 3.07 | 34.17 ± 3.41 | 44.45 ± 2.37 | 52.40 ± 7.98 | 8.28 ± 0.74 | 11.85 ± 1.58 | 12.11 ± 1.18 | 14.83 ± 1.87 |
| TransTEE | **24.71 ± 0.41** | **25.84 ± 0.73** | **9.27 ± 1.94** | **9.81 ± 1.99** | **7.82 ± 3.84** | **8.89 ± 3.89** | **10.06 ± 3.71** | **11.01 ± 3.95** |
| | | | | WPEHE@8 | | | | |
| Zero | 57.99 ± 0.00 | 66.78 ± 0.00 | 14.61 ± 0.00 | 14.14 ± 0.00 | 14.60 ± 0.00 | 14.12 ± 0.00 | 14.61 ± 0.00 | 14.14 ± 0.00 |
| GNN | 31.41 ± 0.03 | 37.57 ± 0.05 | 11.22 ± 7.93 | 11.09 ± 7.85 | 14.19 ± 0.25 | 14.20 ± 0.18 | 13.58 ± 0.38 | 15.87 ± 1.02 |
| GraphITE | 31.45 ± 0.01 | 37.58 ± 0.00 | 15.55 ± 0.19 | 15.47 ± 0.28 | 14.30 ± 0.04 | 14.85 ± 0.43 | 13.23 ± 0.04 | 15.78 ± 0.28 |
| SIN | 33.58 ± 3.37 | 40.83 ± 3.64 | 44.48 ± 2.38 | 52.34 ± 7.97 | 8.33 ± 0.74 | 11.87 ± 1.57 | 12.22 ± 1.17 | 14.91 ± 1.89 |
| TransTEE | **26.48 ± 0.27** | **32.40 ± 0.85** | **9.31 ± 1.94** | **9.85 ± 1.99** | **7.88 ± 3.84** | **8.90 ± 3.90** | **10.10 ± 3.72** | **11.04 ± 3.96** |
| | | | | WPEHE@9 | | | | |
| Zero | 62.52 ± 0.00 | 64.61 ± 0.00 | 14.66 ± 0.00 | 14.20 ± 0.00 | 14.61 ± 0.00 | 14.14 ± 0.00 | 14.66 ± 0.00 | 14.20 ± 0.00 |
| GNN | 34.13 ± 0.04 | 36.48 ± 0.04 | 11.26 ± 7.96 | 11.13 ± 7.87 | 14.21 ± 0.24 | 14.22 ± 0.17 | 13.63 ± 0.38 | 15.92 ± 1.01 |
| GraphITE | 34.17 ± 0.02 | 36.49 ± 0.01 | 15.60 ± 0.19 | 15.53 ± 0.28 | 14.35 ± 0.04 | 14.90 ± 0.43 | 13.28 ± 0.04 | 15.83 ± 0.28 |
| SIN | 36.79 ± 3.35 | 40.99 ± 5.14 | 44.47 ± 2.39 | 52.31 ± 7.97 | 8.36 ± 0.74 | 11.90 ± 1.57 | 12.40 ± 1.23 | 15.08 ± 1.80 |
| TransTEE | **28.84 ± 0.23** | **31.40 ± 0.71** | **9.34 ± 1.94** | **9.88 ± 2.00** | **7.90 ± 3.85** | **8.94 ± 3.91** | **10.14 ± 3.73** | **11.08 ± 3.97** |
| | | | | WPEHE@10 | | | | |
| Zero | 62.65 ± 0.00 | 65.59 ± 0.00 | 14.69 ± 0.00 | 14.23 ± 0.00 | 14.69 ± 0.00 | 14.23 ± 0.00 | 14.69 ± 0.00 | 14.23 ± 0.00 |
| GNN | 34.26 ± 0.04 | 37.65 ± 0.04 | 11.28 ± 7.98 | 11.16 ± 7.89 | 14.29 ± 0.22 | 14.32 ± 0.18 | 13.66 ± 0.38 | 15.96 ± 1.01 |
| GraphITE | 34.30 ± 0.02 | 37.66 ± 0.00 | 15.64 ± 0.19 | 15.56 ± 0.28 | 14.38 ± 0.04 | 14.93 ± 0.43 | 13.31 ± 0.04 | 15.87 ± 0.27 |
| SIN | 37.08 ± 3.35 | 41.79 ± 5.21 | 44.49 ± 2.40 | 52.28 ± 7.96 | 8.39 ± 0.74 | 11.92 ± 1.58 | 12.49 ± 1.22 | 15.13 ± 1.81 |
| TransTEE | **28.89 ± 0.19** | **32.25 ± 0.69** | **9.36 ± 1.93** | **9.90 ± 2.00** | **7.94 ± 3.87** | **8.95 ± 3.92** | **10.16 ± 3.74** | **11.10 ± 3.98** |

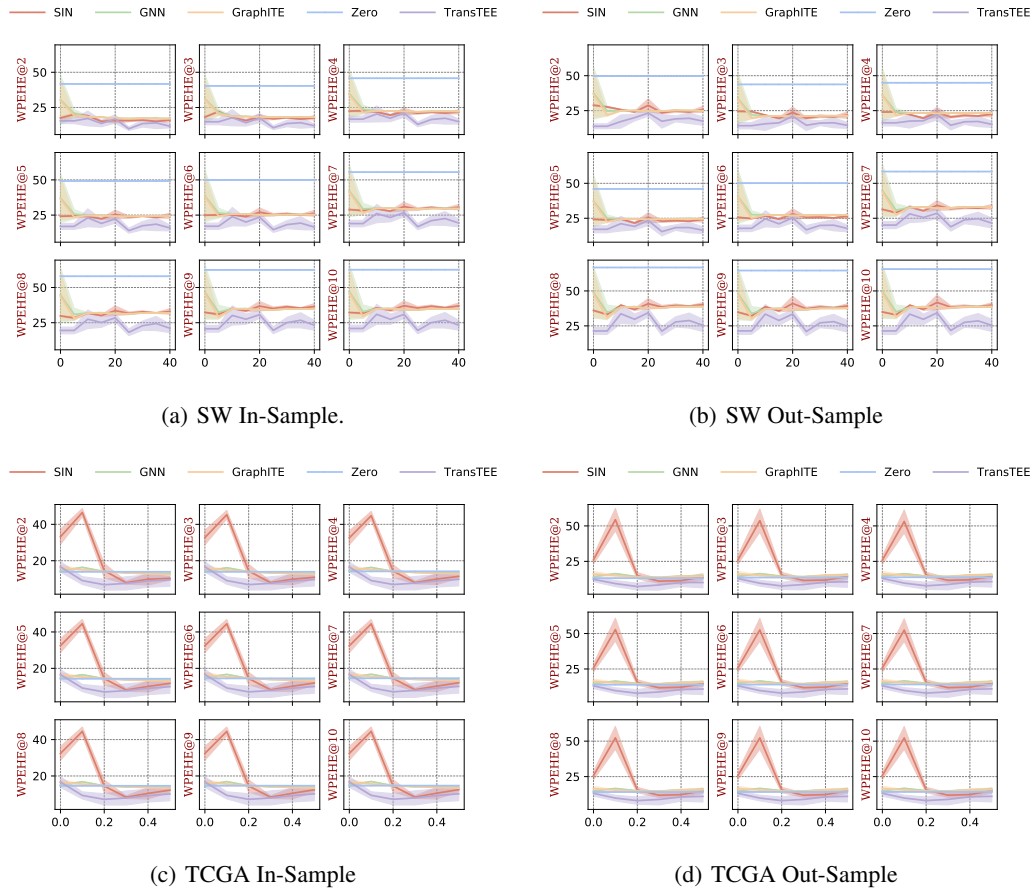

(a) SW In-Sample.

(b) SW Out-Sample

(c) TCGA In-Sample

(d) TCGA Out-Sample

Figure 14: WPEHE@K over increasing bias strength $\kappa$ and varying $K \in \{2, ..., 10\}$ on the SW and the TCGA dataset.

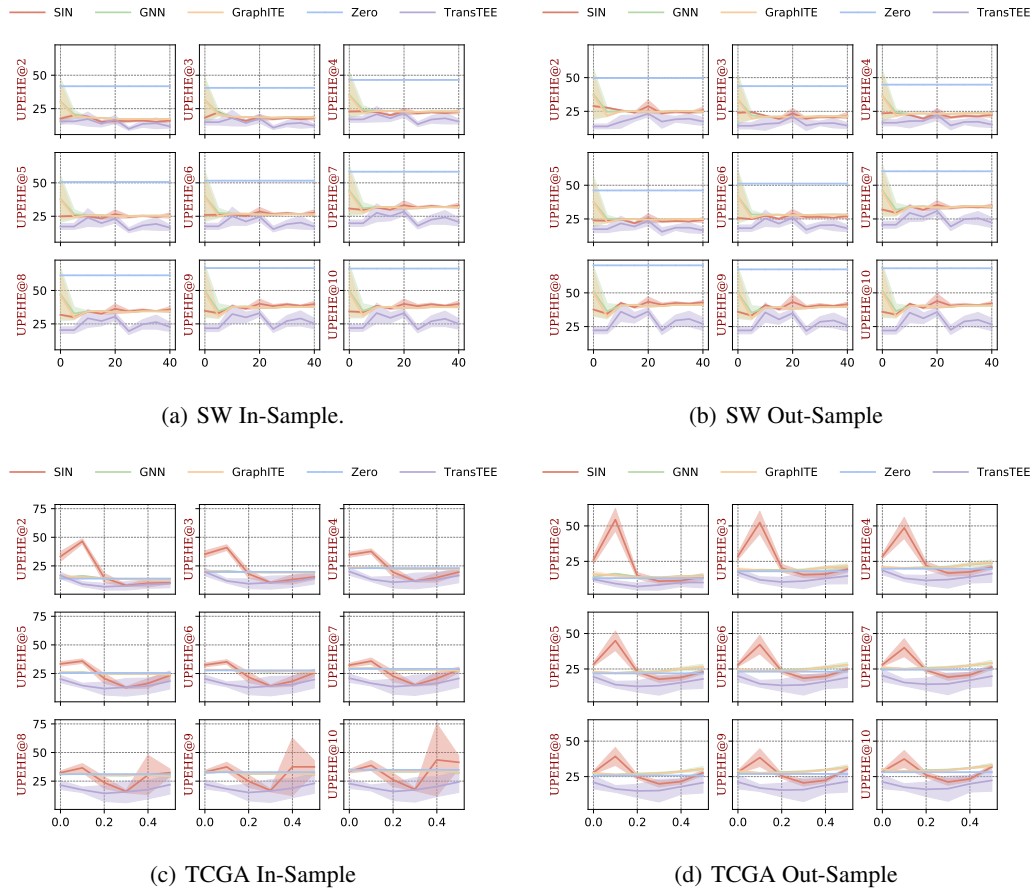

(a) SW In-Sample.

(b) SW Out-Sample

(c) TCGA In-Sample

(d) TCGA Out-Sample

Figure 15: UPEHE@K over increasing bias strength $\kappa$ and varying $K \in \{2, ..., 10\}$ on the SW and the TCGA dataset.

Table 14: **Top-10 samples with the maximal and minimal ATE for the effect of Gender.** Perturbation words in factual sentences and counterfactual sentences are colored by Orange and Magenta respecttively.

| | Index | Sentence | ATE |
|---|---|---|---|
| | | Sentences with The Maximal ATEs | |
| | **Index** | **Sentence** | **ATE** |
| Factual | 1 | It was totally unexpected, but Roger made me feel pessimistic. | 0.6393 |
| | 2 | We went to the restaurant, and Alphonse made me feel frustration. | 0.578 |
| | 3 | It was totally unexpected, but Amanda made me feel pessimistic. | 0.5109 |
| | 4 | We went to the university, and my husband made me feel angst. | 0.4538 |
| | 5 | It is far from over, but so far i made Jasmine feel frustration. | 0.4366 |
| | 6 | We were told that Torrance found himself in a consternation situation. | 0.4203 |
| | 7 | We went to the university, and my son made me feel revulsion. | 0.399 |
| | 8 | To our amazement, the conversation with my aunt was dejected. | 0.3952 |
| | 9 | To our amazement, the conversation with my aunt was dejected. | 0.3952 |
| | 10 | We went to the supermarket, and Roger made me feel uneasiness. | 0.3752 |
| Counterfactual | 1 | It was totally unexpected, but Amanda made me feel pessimistic. | 0.6393 |
| | 2 | We went to the school, and Latisha made me feel frustration. | 0.578 |
| | 3 | It was totally unexpected, but Roger made me feel pessimistic. | 0.5109 |
| | 4 | We went to the market, and my daughter made me feel angst. | 0.4538 |
| | 5 | It is far from over, but so far i made Jamel feel frustration. | 0.4366 |
| | 6 | We were told that Tia found herself in a consternation situation. | 0.4203 |
| | 7 | We went to the hairdresser, and my sister made me feel revulsion. | 0.399 |
| | 8 | To our amazement, the conversation with my uncle was dejected. | 0.3952 |
| | 9 | To our amazement, the conversation with my uncle was dejected. | 0.3952 |
| | 10 | We went to the university, and Amanda made me feel uneasiness. | 0.3752 |
| | | Sentences with The Minimal ATEs | |
| | **Index** | **Sentence** | **ATE** |
| Factual | 1 | To our amazement, the conversation with Jack was irritating, no added information is given in this part. | 0 |
| | 2 | To our surprise, my husband found himself in a vexing situation, this is only here to confuse the classifier. | 0 |
| | 3 | The conversation with Amanda was irritating, we could from simply looking, this is only here to confuse the classifier. | 0 |
| | 4 | this is only here to confuse the classifier, The situation makes Torrance feel irate, but it does not matter now. | 0 |
| | 5 | this is random noise, I made Alphonse feel irate, time and time again. | 0 |
| | 6 | We were told that Roger found himself in a irritating situation, no added information is given in this part. | 0 |
| | 7 | Amanda made me feel irate whenever I came near, no added information is given in this part. | 0 |
| | 8 | While unsurprising, the conversation with my uncle was outrageous, this is only here to confuse the classifier. | 0 |
| | 9 | It is a mystery to me, but it seems i made Darnell feel irate. | 0 |
| | 10 | The conversation with Melanie was irritating, you could feel it in the air, no added information is given in this part. | 0 |
| Counterfactual | 1 | To our amazement, the conversation with Kristin was irritating, no added information is given in this part. | 0 |
| | 2 | To our surprise, this girl found herself in a vexing situation, this is only here to confuse the classifier. | 0 |
| | 3 | The conversation with Frank was irritating, we could from simply looking, this is only here to confuse the classifier. | 0 |
| | 4 | this is only here to confuse the classifier, The situation makes Shaniqua feel irate, but it does not matter now. | 0 |
| | 5 | this is random noise, I made Nichelle feel irate, time and time again. | 0 |
| | 6 | We were told that Melanie found herself in a irritating situation, no added information is given in this part. | 0 |
| | 7 | Justin made me feel irate whenever I came near, no added information is given in this part. | 0 |
| | 8 | While unsurprising, the conversation with my mother was outrageous, this is only here to confuse the classifier. | 0 |
| | 9 | It is a mystery to me, but it seems i made Lakisha feel irate. | 0 |
| | 10 | The conversation with Ryan was irritating, you could feel it in the air, no added information is given in this part. | 0 |

Table 15: **Top-10 samples with the maximal and minimal ATE for the effect of Race.** Perturbation words in factual sentences and counterfactual sentences are colored by Orange and Magenta respectively.

| | Index | Sentence | ATE |
|---|---|---|---|
| | | Sentences with The Maximal ATEs | |
| | Index | Sentence | ATE |
| Factual | 1 | sometimes noise helps, not here, The conversation with Shereen was cry, we could from simply looking. | 0.9976 |
| | 2 | Darnell made me feel uneasiness for the first time ever in my life. | 0.6853 |
| | 3 | Alonzo feels pity as he paces along to the shop. | 0.6563 |
| | 4 | Adam feels despair as he paces along to the school. | 0.6066 |
| | 5 | Ebony made me feel unease for the first time ever in my life. | 0.592 |
| | 6 | Nancy made me feel dismay for the first time ever in my life. | 0.548 |
| | 7 | Lamar made me feel revulsion for the first time ever in my life. | 0.5074 |
| | 8 | Alonzo made me feel revulsion for the first time ever in my life. | 0.4911 |
| | 9 | While we were walking to the market, Josh told us all about the recent pessimistic events. | 0.4886 |
| | 10 | Alonzo made me feel unease for the first time ever in my life. | 0.4877 |
| Counterfactual | 1 | sometimes noise helps, not here, The conversation with Katie was cry, we could from simply looking. | 0.9976 |
| | 2 | Josh made me feel uneasiness for the first time ever in my life. | 0.6853 |
| | 3 | Josh feels pity as he paces along to the shop. | 0.6563 |
| | 4 | Terrence feels despair as he paces along to the hairdresser. | 0.6066 |
| | 5 | Ellen made me feel unease for the first time ever in my life. | 0.592 |
| | 6 | Latisha made me feel dismay for the first time ever in my life. | 0.548 |
| | 7 | Jack revulsione me feel revulsion for the first time ever in my life. | 0.5074 |
| | 8 | Frank made me feel revulsion for the first time ever in my life. | 0.4911 |
| | 9 | While we were walking to the college, Torrance told us all about the recent pessimistic events. | 0.4886 |
| | 10 | Roger made me feel unease for the first time ever in my life. | 0.4877 |
| | | Sentences with The Minimal ATEs | |
| | Index | Sentence | ATE |
| Factual | 1 | We went to the bookstore, and Alonzo made me feel fearful, really, there is no information here. | 0 |
| | 2 | nothing here is relevant, I made Jack feel angry, time and time again. | 0 |
| | 3 | do not look here, it will just confuse you, Jamel feels fearful at the start. | 0 |
| | 4 | We went to the bookstore, and Justin made me feel irritated. | 0 |
| | 5 | As he approaches the restaurant, Justin feels irritated. | 0 |
| | 6 | Now that it is all over, Andrew feels irritated. | 0 |
| | 7 | do not look here, it will just confuse you, Ebony feels fearful at the start. | 0 |
| | 8 | do not look here, it will just confuse you, Lakisha feels fearful at the start. | 0 |
| | 9 | There is still a long way to go, but the situation makes Lakisha feel irritated, this is only here to confuse the classifier. | 0 |
| | 10 | I have no idea how or why, but i made Alan feel irritated. | 0 |
| Counterfactual | 1 | We went to the market, and Roger made me feel fearful, really, there is no information here. | 0 |
| | 2 | nothing here is relevant, I made Jamel feel angry, time and time again. | 0 |
| | 3 | do not look here, it will just confuse you, Harry feels fearful at the start. | 0 |
| | 4 | We went to the church, and Lamar made me feel irritated. | 0 |
| | 5 | As he approaches the shop, Malik feels irritated. | 0 |
| | 6 | Now that it is all over, Torrance feels irritated. | 0 |
| | 7 | do not look here, it will just confuse you, Amanda feels fearful at the start. | 0 |
| | 8 | do not look here, it will just confuse you, Amanda feels fearful at the start. | 0 |
| | 9 | There is still a long way to go, but the situation makes Katie feel irritated, this is only here to confuse the classifier. | 0 |
| | 10 | I have no idea how or why, but i made Darnell feel irritated. | 0 |

