# OpenReview forum: "Exploring Transformer Backbones for Heterogeneous Treatment Effect Estimation"
_ICLR.cc/2023/Conference — Submitted to ICLR 2023_

### Official Review · Reviewer_NSTZ · 2022-10-24

**Confidence:** 3
**Correctness:** 4
**Technical Novelty And Significance:** 3
**Empirical Novelty And Significance:** 2
**Recommendation:** 6

**Clarity, Quality, Novelty And Reproducibility:**

This paper is clearly written and high-quality, but slightly lacks novelty.
This is not the first transformer-based paper I've seen in the causal field.


**Strength And Weaknesses:**

Strength.
1.	This paper verifies the effectiveness of the transformer for treatment effect estimation tasks, which is important for causality. And they conducted many experiments to demonstrate the effectiveness of the proposed transformer.
2.	They analyze in detail the essential properties of the proposed method in causal: compatibility with propensity score modeling, parameter efficiency, robustness to continuous treatment value distribution shifts, and explainable in covariate adjustment.

Weaknesses.
The proposed transformer lacks novelty in the methodology.


**Summary Of The Paper:**

This paper carefully designs a transformer as the backbone for treatment effect estimation, which is applicable to various tasks. They conduct many experiments to demonstrate the effectiveness of the proposed backbone and analyze the properties of the proposed transformer based on these results.

**Summary Of The Review:**

This paper conducts extensive experiments to verify the effectiveness of transformer. This paper demonstrates that transformer has many good properties in the causal domain. From the paper's perspective, the approach proposed in this paper lacks a certain degree of novelty.

---

> ### Author Response · Authors · 2022-11-14
> **Response to Review  NSTZ**
>
> We sincerely thank you for your helpful comments. The technical novelty is also raised by other reviewers and please see **General Response** for more details. We make the following comparison to existing Transformer based methods both conceptually and empirically.
>
> In conclusion, in addition to the differences discussed in the Related Work section, the difference lies in two categories.
>
> 1. **The main design choices of TransTEE are different from existing methods**, which results in a substantial performance improvement on various TEE settings.
>
> 2. **TransTEE can be applied to most existing heterogeneous TEE tasks**, however, existing transformer based methods [1,2] can only work well on some specific scenarios, such as discrete or just one treatment.
>
> For details, compared to existing transformer-based papers in the causal field,  CETransformer and ANU, we have the following differences.
>
> CETransformer[1] embeds covariates for different treatments as a T-learner. They only trivially learn covariate embeddings but not treatment embedding, while the latter is shown more important for TEE tasks. In contrast, TransTEE is an S-learner, which is more well-suited to account for causal heterogeneity
>
> ANU[2] is more related to the proposed TransTEE. There are two main differences between these two methods. (i) The model structure is different. ANU performs cross-attention between covariates and treatments while no self-attention is applied. However, self-attention layers are of importance for high-dimensional data (see the following experiments). The proposed TransTEE uses self-attention to process covariates and treatments respectively, which attains superior performance. (ii) Different applicable scenarios. ANU cannot be applied to multi-treatment settings, which have been extensively studied recently[3,4]. In contrast, TransTEE can almost be applied to all existing heterogeneous settings.
>
> We compare the attentive neural uplift model (ANU) [2] and CETransformer[1]  with ours in the following two settings. (1) IHDP dataset in Table.2 in the main manuscript. We adjust the layers of CETransformer and ANU such that the total parameters of  these models are similar. The result is shown in the following table. With the usage of treatment embeddings, ANU is shown to be more robust than VCNet and DRNet when a treatment shift occurs. CETransformer uses different prediction branches for different treatment groups, which is similar to DRNet, which results in inferior performance when treatments are continuous or out of the training distribution. In both the binary treatment setting and continuous treatment settings, TransTEE performs better than ANU and CETransformer.
>
>
> |  Methods |   VANILLA (BINARY)   |   VANILLA (h = 1)   | EXTRAPOLATION (h = 2) |
> |:--------:|:--------------------:|:-------------------:|:---------------------:|
> |   DRNet  |    0.3543 ± 0.6062   |   2.1549 ± 1.04483  |    11.071 ± 0.9938    |
> |   VCNet  |   0.2098 ± 0.18236   |   0.7800 ± 0.6148   |          NAN          |
> |   CETransformer[1] |   0.2317 ± 0.11237   |   1.3217 ± 0.2981   |          5.8281 ± 0.29183        |
> |  ANU [2] |   0.1482 ± 0.17362   |   0.2147± 0.32451   |    0.4244 ± 0.19832   |
> | TransTEE | **0.0983 ± 0.15384** | **0.1151 ± 0.1028** |  **0.2745 ± 0.1497**  |
>
> (2) We further evaluate the real-world utility of ANU [2] and the experimental setting is detailed in Section 5.4 in the main paper. Covariates here are long sentences. Thanks to the use of self-attention modules, TransTEE can achieve better estimation results compared to baselines. For ANU, no self-attention layer is applied, and the final estimation is inaccurate, which verifies the superiority of the proposed framework.
>
> |        | Correlation/Representation Based Baselines |        |        |       |             | Treatment Effect Estimators |        |             |            |
> |:------:|:------------------------------------------:|:------:|:------:|:-----:|:-----------:|:---------------------------:|:------:|:-----------:|:----------:|
> |   TC   |                 ATE$_{GT}$                 | TReATE | CONEXP |  INLP | CausalBERT  |            TarNet           |  DRNet| CETransformer[1]  | **ANU [2]** |  TransTEE  |
> | Gender |                    0.086                   |  0.125 |  0.02  | 0.313 |    0.179    |            0.0067           | 0.0088 | 0.0096 |   0.0042    |  **0.013** |
> |  Race  |                    0.014                   |  0.046 |  0.08  | 0.591 |    0.213    |            0.005            |  0.006 | 0.0075|   0.0083    | **0.0174** |
>
> The above discussion has been included in our draft. Thanks for your kind suggestion again.
>
>
> [1]  ``CETransformer: Casual Effect Estimation via Transformer Based Representation Learning." PRCV, 2021.
>
> [2]  "Learning Discriminative Representation Based on Attention for Uplift." PAKDD, 2022.

---

> ### Author Response · Authors · 2022-11-28
> **A Gentle Reminder**
>
> Dear Reviewer NSTZ,
>
> Thanks again for your careful reading and valuable comments to improve our submission. We want to leave a gentle reminder due to the closing end time of the discussion period.
>
> We have tried our best to address your concerns point by point with detailed explanations and results. Moreover, the draft was revised correspondingly. We would really appreciate your feedback to make sure the responses and revisions have addressed all your concerns, or whether there is a leftover concern we can address.
>
> Sincerely
>
> Authors of Paper2191

---

### Official Review · Reviewer_u5ZN · 2022-10-25

**Confidence:** 5
**Clarity, Quality, Novelty And Reproducibility:** 1. The paper is well-written and easy…
**Correctness:** 3
**Technical Novelty And Significance:** 2
**Empirical Novelty And Significance:** 3
**Recommendation:** 5

**Strength And Weaknesses:**

Strength:
The studied problem is very important in practice and the paper is well-written. The paper is a comprehensive study that explores the effectiveness and efficiency of Transformers in TEE problems.

Weaknesses:
1.  The technical contribution of this paper is not very solid.
	a. The Transformers are well-studied and show superior performance over other neural models (MLPs, CNN or RNN) in many domains including the TEE (e.g., CETransformer and ANU). It is not very surprising that the proposed Transformer-based method can also outperform the baselines in TEE problems.
	b. The treatment prediction head is trained in an adversarial pattern to alleviate the treatment bias which is already been used in many existing works (e.g., Bica et al., 2020; Kallus 2020).

2. For the covariate adjustment via cross-attention module, the learned attention weights tend to be very unstable and thus unreliable to be used for covariate selection and interpretability. As shown in Table 4, different variations of the proposed model yields different and inconsistent results in learned attention weights. The results will become more unstable in real-world scenarios with high dimensional covariate space and complex relationships among the covariates.

3. The motivation behind using separated linear layers for treatment and dosage is to alleviate the issue of treatment distribution shift in the test (i.e., some treatment values in the test are not seen during training). However, whether this motivation and experiment design will violate the positivity assumption that the probability of receiving any treatment is non-zero? For example, in extrapolation (h=5), t\in[0.25, 5.0] in training and t\in[0, 5] in testing. Thus the individuals in the training set will have zero probability to receive t\in[0, 0.25).  Then the problem becomes whether it is meaningful to study the treatment distribution shift problem in TEE under the positivity assumption.

4. It would be great to incorporate the comparison (from conceptual-level and empirical analysis) with existing Transformer-based TEE methods in the main paper. These methods should be the most related baselines to compare with.

Bica, Ioana, et al. "Estimating counterfactual treatment outcomes over time through adversarially balanced representations." ICLR 2020.
Kallus, Nathan. "Deepmatch: Balancing deep covariate representations for causal inference using adversarial training." ICML 2020.


**Summary Of The Paper:**

The authors propose to adopt Transformers as model backbones for treatment effect estimation. Comprehensive experiments are conducted to show the effectiveness of the proposed Transformer-based TEE methods over the existing MLP-based TEE methods.

**Summary Of The Review:**

The studied problem is very important in practice and the paper is well-written. It is a comprehensive paper, but not a technically novel one. There are also some technical questions in the experiments.

---

> ### Author Response · Authors · 2022-11-14
> **Response to Reviewer u5ZN [1/2]**
>
> We thank the reviewer for the careful reading and valuable feedback and hope our response could resolve your concerns.
>
> **[Concern 1]. The technical contribution of this paper is not very solid.**
>
> Please see **General Response** for more details.
>
> *Q1.a:  The Transformers are well-studied for TEE tasks.*
>
> The previous work ((e.g., CETransformer [1] and ANU [2])) did make some attempts to use the transformer framework and we have cited and discussed all these methods (See Appendix A).
>
> From the conceptual level, compared to CETransformer and ANU, we have the following differences.
>
> CETransformer [1] embeds covariates for different treatments as a T-learner. They only trivially learn covariate embeddings but not treatment embedding, while the latter is shown more important for TEE tasks. In contrast, TransTEE is an S-learner, which is more well-suited to account for causal heterogeneity as shown in [6].
>
> ANU [2] is more related to the proposed TransTEE. There are two main differences between these two methods. (i) The model structure is different. ANU performs cross-attention between covariates and treatments while no self-attention is applied. However, self-attention layers are of importance for high-dimensional data (see the following experiments). The proposed TransTEE uses self-attention to process covariates and treatments respectively, which attains superior performance. (ii) Different applicable scenarios. ANU cannot be applied to multi-treatment settings, which have been extensively studied recently [3,4]. In contrast, TransTEE can be more flexible and applicable for a wide range of settings.
>
> We compare the attentive neural uplift model (ANU) [2] and CETransformer [1]  with ours in the following two settings. (1) IHDP dataset in Table.2 in the main manuscript. We adjust the layers of CETransformer and ANU such that the total parameters of these models are similar. The result is shown in the following table. With the usage of treatment embeddings, ANU is shown to be more robust than VCNet and DRNet when a treatment shift occurs. CETransformer uses different prediction branches for different treatment groups, which is similar to DRNet, which results in inferior performance when treatments are continuous or out of training distribution. In both the binary treatment and continuous treatment settings, TransTEE outperforms ANU and CETransformer.
>
>
> |  Methods |   VANILLA (BINARY)   |   VANILLA (h = 1)   | EXTRAPOLATION (h = 2) |
> |:--------:|:--------------------:|:-------------------:|:---------------------:|
> |   DRNet  |    0.3543 ± 0.6062   |   2.1549 ± 1.04483  |    11.071 ± 0.9938    |
> |   VCNet  |   0.2098 ± 0.18236   |   0.7800 ± 0.6148   |          NAN          |
> |   CETransformer[1] |   0.2317 ± 0.11237   |   1.3217 ± 0.2981   |          5.8281 ± 0.29183        |
> |  ANU [2] |   0.1482 ± 0.17362   |   0.2147± 0.32451   |    0.4244 ± 0.19832   |
> | TransTEE | **0.0983 ± 0.15384** | **0.1151 ± 0.1028** |  **0.2745 ± 0.1497**  |
>
> (2) We further evaluate the real-world utility of ANU [2] and the experimental setting is detailed in Section 5.4 in the main paper. Covariates here are long sentences. Thanks to the use of self-attention modules, TransTEE can achieve better estimation results compared to baselines. For ANU, no self-attention layer is applied, and the final estimation is inaccurate, which verifies the superiority of the proposed framework.
>
> |        | Correlation/Representation Based Baselines |        |        |       |             | Treatment Effect Estimators |        |             |            |
> |:------:|:------------------------------------------:|:------:|:------:|:-----:|:-----------:|:---------------------------:|:------:|:-----------:|:----------:|
> |   TC   |                 ATE$_{GT}$                 | TReATE | CONEXP |  INLP | CausalBERT  |            TarNet           |  DRNet| CETransformer[1]  | **ANU [2]** |  TransTEE  |
> | Gender |                    0.086                   |  0.125 |  0.02  | 0.313 |    0.179    |            0.0067           | 0.0088 | 0.0096 |   0.0042    |  **0.013** |
> |  Race  |                    0.014                   |  0.046 |  0.08  | 0.591 |    0.213    |            0.005            |  0.006 | 0.0075|   0.0083    | **0.0174** |
>
> [1]  `CETransformer: Casual Effect Estimation via Transformer Based Representation Learning. PRCV 2021.
>
> [2]  Learning Discriminative Representation Based on Attention for Uplift." Pacific-Asia Conference on Knowledge Discovery and Data Mining. Springer, Cham, 2022.
>
> [3] Estimating the effects of continuous-valued interventions using generative adversarial networks. Advances in Neural Information Processing Systems, 2020.
>
> [4] Ncore: Neural counterfactual representation learning for combinations of treatments. arXiv 2021.
>
> [5] Deepmatch: Balancing deep covariate representations for causal inference using adversarial training. ICML 2020.
>
> [6] On Inductive Biases for Heterogeneous Treatment Effect Estimation. NeurIPS 2021

---

> > ### Author Response · Authors · 2022-11-14
> > **Response to Reviewer u5ZN [2/2]**
> >
> > *Q1.b:  Adversarial training to alleviate the treatment bias.*
> >
> > Please note that the use of adversarial training is only one of our contributions. And we have discussed the differences between [3,5] and TR/PTR in the paper: most works like [3,5] use adversarial training only for binary treatments while TR/PTR are used for continuous treatments and have specific theoretical properties (they match the first and second moments of $p(\Phi_x(x))$ and $p(\Phi_t(t)|\Phi_x(x))$.
> > At the very least, conducting experiments to show the effectiveness of TransTEE when a propensity score model is learned can verify the compatibility of TransTEE to common techniques in causal inference so that users can have broader use cases.
> >
> > ----
> >
> > **[Concern 2]. The learned attention weights tend to be very unstable and thus unreliable to be used for covariate selection and interpretability.**
> >
> > Thanks for your helpful suggestions, we make the following clarification.
> >
> > - Firstly, we admit that the attention weights can be unstable and can vary across different experimental random seeds. However, it is the first attempt to explain the internal decision-making process of TEE tasks.
> > - Besides, we do our best to eliminate the variance of the experiment, as shown in the experimental section, for all experiments about attention weights, we conduct 10 repetitions for TransTEE and its TR, PTR counterparts over 100 datasets, the attention weights are averaged from all datasets. The difference of attention weights between TransTEE and its TR, PTR counterparts is not unstable, which in contrast verifies the compatibility of TransREE with propensity score modeling. Specifically, TransTEE assigns more weights to confounding covariates and fewer weights on noisy covariates, while both TR and PTR improve confounding control, namely both TR and PTR assign more weights to confounders.
> >
> > We attach the *standard deviation* of the three groups of attention weights as follows, where the std is not large with a large number of datasets and repetitions.
> >
> > | covariates   | $w_{con}$ | w_1    | w_2    |
> > |--------------|-----------|--------|--------|
> > | TransTEE     |    0.1070 | 0.0581 | 0.0592 |
> > | TransTEE+TR  |    0.0529 | 0.0548 | 0.0730 |
> > | TransTEE+PTR |    0.0687 | 0.0570 | 0.0491 |
> >
> > ----
> >
> > **[Concern 3]. whether it is meaningful to study the treatment distribution shift problem in TEE under the positivity assumption.**
> >
> > Thanks for pointing this out. We updated the draft to avoid confusion.
> >
> > We believe that it does not violate the positivity assumption, as by definition, positivity only gives constraints on training data.
> > Recall that positivity gives us $0<\pi(t \mid x)<1, \forall x \in \mathcal{X}, t \in \mathcal{T}$. And $\mathcal{T}$ denotes the training treatment data distribution ([0.25, 5.0] as you mentioned). And deployment to out-of-distribution treatment settings can be independent of the positivity assumption on the training data.
> >
> >
> > ----
> >
> > **[Concern 4]. It would be great to incorporate the comparison (from conceptual-level and empirical analysis) with existing Transformer-based TEE methods in the main paper.**
> >
> > Thanks for the valuable suggestion. We will include the experimental comparison results above and move the conceptual-level comparison from the Appendix to the main paper per your suggestion.
> >
> >
> > [1] CETransformer: Casual Effect Estimation via Transformer Based Representation Learning. PRCV, 2021.
> >
> > [2] Learning Discriminative Representation Based on Attention for Uplift. Pacific-Asia Conference on Knowledge Discovery and Data Mining. Springer, Cham, 2022.
> >
> > [3] Estimating the effects of continuous-valued interventions using generative adversarial networks. NeurIPS, 2020.
> >
> > [4] Ncore: Neural counterfactual representation learning for combinations of treatments. arXiv 2021.
> >
> > [5] Deepmatch: Balancing deep covariate representations for causal inference using adversarial training. ICML 2020.

---

> ### Author Response · Authors · 2022-11-28
> **A Gentle Reminder**
>
> Dear Reviewer u5ZN,
>
>
> Thanks again for your careful reading and valuable comments to improve our submission. We want to leave a gentle reminder due to the closing end time of the discussion period.
>
> We have tried our best to address your concerns point by point with detailed explanations and results. Moreover, the draft was revised correspondingly. We would really appreciate your feedback to make sure the responses and revisions have addressed all your concerns, or whether there is a leftover concern we can address.
>
> Sincerely
>
> Authors of Paper2191

---

### Official Review · Reviewer_mZnE · 2022-10-26

**Confidence:** 4
**Correctness:** 3
**Technical Novelty And Significance:** 3
**Empirical Novelty And Significance:** 3
**Recommendation:** 6

**Clarity, Quality, Novelty And Reproducibility:**

Clarity

The paper is clear in its assumptions and does a good job of verifying its hypothesis.

Novelty

The paper presents a novel approach to the problem of interest and highlights the differences compared to previous works.

Reproducibility

One may find it challenging to reproduce the results, given the authors did not publish their source code.

**Strength And Weaknesses:**

Strengths:

- The authors used recent advances in NN architectures (e.g., transformers) to design a new method to tackle the Treatment Effect Estimation problem. For instance, they propose different mappings for covariates and treatments, which avoid missing information about treatments given their lower dimensional representation.
- The authors considered different modalities of treatments. Namely, treatment with binary/continuous dosages, structured treatments, and language data. The broad of distinct treatment types helps assess their method's applicability.

Weaknesses:

- The assumption related to unconfoundeness is a very limiting factor in the usage of the proposed method in real-world applications.
- The authors provide extensive analysis using different datasets. However, the reproducibility of their results is difficult, given that they did not make the source code available.
- The authors should clarify the T- and S-learner approaches to causal inference. Most readers are not familiar with these terms.

Minor suggestions:

- It seems to have a typo in Assumption 2: " such that, i.e."  -> "i.e.,"

**Summary Of The Paper:**

This paper is interested in estimating the heterogeneous treatment effect (TEE). The authors argued that TEE is not used in practice because current approaches to tackling it make solid parametric assumptions.

According to the authors, the limitations of previously proposed approaches mainly relate to their poor generalizability. Hence, they tend to focus on a narrower context and ignore other scenarios.

The method present in this paper is based on transformers and their backbone modules. Specifically, its main module is the attention layer.

My main concern in this paper is the assumption about unconfoundeness, which states that there are no hidden confounders. Since the authors are arguing against the applicability of previous works, this assumption impedes using their approach in real applications as well.

**Summary Of The Review:**

This paper proposes to advance the literature by leveraging recent developments on NNs to tackle the Treatment Effect Estimation problem. They provide extensive experiments and detailed information about their proposed method. However, they argued that previous approaches have issues with real-world applicability. When they ignore the presence of confounders, this paper seems to suffer from the same applicability problem.

---

> ### Author Response · Authors · 2022-11-14
> **Response to Reviewer mZnE**
>
> We thank the reviewer for the careful reading and valuable feedback on our submission and hope our response could solve your concerns.
>
> ----
>
> Thank you for the helpful comments.
>
> **[concern 1]The unconfoundedness assumption.**
>
> We agree that relaxing the unconfoundedness is an interesting open research problem.
> But we argue that it is a fundamental and basic property that is assumed by most causal inference methods. For example, all the works in Tab. 1 and other dominant causal inference methods like causal forest and meta-learners all assume unconfoundedness.
> Despite the strong assumption, in practice, those methods are still widely used and please find more practical use cases in libraries like DoWhy (https://github.com/py-why/dowhy) and EconML (https://github.com/microsoft/EconML).
> Please also note that we’ve noted in the Conclusion section that practitioners should pay attention to incorporating sufficiently rich covariates for better suiting their methods.
>
> [1] Estimation and Inference of Heterogeneous Treatment Effects using Random Forests
>
> ----
>
> **[concern 2] Reproducibility.**
>
> Thanks for pointing this out. Please note that we’ve already included all the codes in the supplementary materials. Due to the anonymity rule of ICLR, we can not post the GitHub link here. But we have open-sourced our implementation for TransTEE through GitHub.
>
> ----
>
> **[concern 3] T- and S-learner.**
>
> Please note that we’ve included the citation [2] and those are basic concepts and common knowledge for causal inference or heterogeneous TEE. For completeness, we formally introduce them below and will update them to our latest draft:
> Consider a binary treatment $T$, a continuous outcome $Y$, and confounding covariates $X$. We wish to estimate the effect of $T$ on $Y$ conditioned on $X$. This is called the conditional average treatment effect (CATE), mathematically defined as $\tau(x)=\mathbb{E}[Y(1)-Y(0) \mid X=x] $.
> S-learner gives $\hat{\mu}=M(Y \sim(X, T))$ and $\hat{\tau}(x)=\hat{\mu}(x, 1)-\hat{\mu}(x, 0)$.
> T-learner gives $\hat{\mu}_0=M\left(Y^\nu \sim X^\nu\right), \hat{\mu}_1=M\left(Y^1 \sim X^1\right), \hat{\tau}(x)=\hat{\mu}_1(x)-\hat{\mu}_0(x)$
>
> [2] Meta-learners for Estimating Heterogeneous Treatment Effects using Machine Learning

---

> ### Author Response · Authors · 2022-11-28
> **A Gentle Reminder**
>
> Dear Reviewer mZnE,
>
> Thanks again for your careful reading and valuable comments to improve our submission. We want to leave a gentle reminder due to the closing end time of the discussion period.
>
> We have tried our best to address your concerns point by point with detailed explanations and results. Moreover, the draft was revised correspondingly. We would really appreciate your feedback to make sure the responses and revisions have addressed all your concerns, or whether there is a leftover concern we can address.
>
> Sincerely
>
> Authors of Paper2191

---

### Author Response · Authors · 2022-11-14
**General Response**

We thank the thorough and insightful comments from all the reviewers. The reviewers noted that our framework is promising, the performance of our proposed model is strong, and our study is helpful to researchers in the fields of causal inference. All three reviewers acknowledge that our empirical study is comprehensive and thorough.

Each individual reviewer further has their own personalized questions about the underlying texts and statements of the paper. We have answered the common questions below and also provide detailed discussions of evaluation results in the individual reviewer comments, as well as additional experiments.

> Novelty and Motivation (Reviewer u5ZN, NSTZ)

Recall that our primary contribution is empirically investigating and exploring the role of transformers-based models in causal inference against the trend of handcrafting domain-specific architectures in the literature as discussed in the introduction. This is motivated by the fact that in causal representation learning models are expected to extract features from raw data using deep neural networks while estimating the variables of interests encoded in causal structures.

In general, we believe **the use of previously developed architecture for new tasks has no causal relationship with the soundness of a paper. And having a broader range of use cases can be a significant contribution even if two works are conceptually similar.** For example, transformers are proposed around 2017 but have only been used for image recognition in 2020 [1] and then got widely extended and improved in 2021 [2]. Specifically, Swin Transformer [2] builds hierarchical feature maps to reduce the computation complexity such that it can thus serve as a general-purpose backbone for various vision tasks including image classification and dense recognition. In comparison, ViT [1] produces feature maps of a single resolution and have quadratic complexity, which limits its utility to the dense vision tasks.

Therefore, we highlight two aspects: firstly, building upon attention mechanisms and having similar components to concurrent works might not mean the insights this work contributes to the community are diminished: transformers can be effective and flexible for a wide range of TEE tasks through a thorough analysis. Secondly, including additional advantages and results of TransTEE might not degrade the overall contributions of our work. And TransTEE is not possible to be perfect in every aspect due to no free lunch, especially when some aspects like interpretability and relaxing unconfoundedness deserve separate papers to discuss.

Moreover, we acknowledge that our work is not the first to use transformers to estimate causal effects. However, we highlight that our contribution lies more broadly in investigating the role and flexibility of transformer-based models in TEE, especially with high-dimensional inputs like molecular graphs, texts, and continuous treatments. On the contrary, the mentioned related works: CETransformer [3] is a T-learner is fundamentally different from ours as an S-learner ; ANU [4] is a concurrent work and designed explicitly for uplift modeling with binary treatment rather than for large-scale empirical studies about the generalizability of transformers.
It is **especially underexplored in the literature of statistics and econometrics literature where TEE is more widely studied in theory based on which neural network architectures are built for each task (like all the models in the Table 1)**.

Practically, **auditing the fair prediction of pre-trained models is also an important use case [5] largely neglected by most recent works**. We believe that those aspects in our paper could help provide fresh insights into the causal inference area.

[1] An Image is Worth 16x16 Words: Transformers for Image Recognition at Scale. ICLR 2021

[2] Swin Transformer: Hierarchical Vision Transformer using Shifted Windows. ICCV 2021

[3] CETransformer: Causal Effect Estimation via Transformer Based Representation Learning. PRCV 2021.

[4] Learning Discriminative Representation Based on Attention for Uplift.

[5] CEBaB: Estimating the Causal Effects of Real-World Concepts on NLP Model Behavior.

---

> Conclusion

We further respond to the questions proposed by each individual reviewer below. Please feel free to let us know if you have more questions about the paper. We will try our best to address your concerns.

---

### Decision · Program_Chairs · 2023-01-20

**Decision:**

Reject

**Justification For Why Not Higher Score:**

Although this paper studies an important problem and presents very interesting results, there are still remaining issues that might not be easily addressed with a minor revision, such as lack of theoretical justifications for the studied problem.

**Justification For Why Not Lower Score:**

N/A

**Metareview: Summary, Strengths And Weaknesses:**

This paper investigates transformers for treatment effect estimation, and provides experimental results to demonstrate the effectiveness of the transformer based methods.

Overall, this paper is well written and easy to follow. Treatment effect estimation is an important problem in causal inference. This paper presents comprehensive evaluations of transformer-based methods and MLP-based methods for treatment effect estimation.

Meanwhile, this paper has some limitations. In particular, the novelty of proposed method is not significant, as transformers and adversarial learning have been widely used for treatment effect estimation. Also, some design choices are not well justified. The authors provided detailed responses as well as additional experimental results, which addressed some of the concerns from reviewers. However, during the discussion phase, reviewers pointed out that there are some remaining issues, such as the theoretical justification of the treatment distribution shift problem under the positivity assumption. Hopefully, the remaining issues could be addressed in the next version of this work.

**Summary Of Ac-Reviewer Meeting:**

N/A